# Suppressor of cytokine signaling (SOCS)5 ameliorates influenza infection via inhibition of EGFR signaling

Lukasz Kedzierski[1,2†], Michelle D Tate[3,4†], Alan C Hsu[5†], Tatiana B Kolesnik[1,2], Edmond M Linossi[1,2], Laura Dagley[1,2], Zhaoguang Dong[6], Sarah Freeman[1,2], Giuseppe Infusini[1,2], Malcolm R Starkey[5], Nicola L Bird[7], Simon M Chatfield[1,2], Jeffrey J Babon[1,2], Nicholas Huntington[1,2], Gabrielle Belz[1,2], Andrew Webb[1,2], Peter AB Wark[5], Nicos A Nicola[1,2], Jianqing Xu[6], Katherine Kedzierska[7], Philip M Hansbro[5*‡], Sandra E Nicholson[1,2*‡]

[1]The Walter and Eliza Hall Institute of Medical Research, Parkville, Australia; [2]Department of Medical Biology, University of Melbourne, Parkville, Australia; [3]Centre for Innate Immunity and Infectious Diseases, Hudson Institute of Medical Research, Clayton, Australia; [4]Monash University, Clayton, Australia; [5]Priority Research Centre for Asthma and Respiratory Diseases, Hunter Medical Research Institute and The University of Newcastle, Newcastle, Australia; [6]Shanghai Public Health Clinical Center and Institutes of Biomedical Sciences, Fudan University, Shanghai, China; [7]Department of Microbiology and Immunology, University of Melbourne, at the Peter Doherty Institute for Infection and Immunity, Parkville, Australia

*For correspondence: philip. hansbro@newcastle.edu.au (PMH); snicholson@wehi.edu.au (SEN)

†These authors contributed equally to this work
‡These authors also contributed equally to this work

Competing interests: The authors declare that no competing interests exist.

**Abstract** Influenza virus infections have a significant impact on global human health. Individuals with suppressed immunity, or suffering from chronic inflammatory conditions such as COPD, are particularly susceptible to influenza. Here we show that suppressor of cytokine signaling (SOCS) five has a pivotal role in restricting influenza A virus in the airway epithelium, through the regulation of epidermal growth factor receptor (EGFR). *Socs5*-deficient mice exhibit heightened disease severity, with increased viral titres and weight loss. *Socs5* levels were differentially regulated in response to distinct influenza viruses (H1N1, H3N2, H5N1 and H11N9) and were reduced in primary epithelial cells from COPD patients, again correlating with increased susceptibility to influenza. Importantly, restoration of SOCS5 levels restricted influenza virus infection, suggesting that manipulating SOCS5 expression and/or SOCS5 targets might be a novel therapeutic approach to influenza.

## Introduction

Influenza A virus is a single stranded RNA virus that infects the upper respiratory tract and has a major impact on global health. In most instances the effects are largely socioeconomic, with infected individuals requiring bed-rest and recovering at home. However, for the very young, pregnant women, the elderly and the infirm, severe influenza can result in hospitalisation and even death, with seasonal strains accounting for 500,000 deaths annually (*Hsu et al., 2012b*; *Vanders et al., 2015*). Those that succumb to the disease can die as a result of an unrestrained inflammatory response (often referred to as a 'cytokine storm'), which together with cell death irretrievably damages the airways releasing fluid into the alveolar spaces (*Short et al., 2014*). Individuals who suffer from pre-

**eLife digest** Influenza, commonly referred to as the flu, is a highly contagious disease caused by a virus. When an infected person coughs or sneezes, droplets containing the virus are released into the air. Other individuals nearby may breathe in the virus, which then enters the cells lining the lungs and multiplies.

Some people are more susceptible to the influenza virus than others. In particular, individuals with chronic obstructive pulmonary disease (COPD) often suffer much worse flu symptoms and are more likely to be admitted to hospital. COPD results from smoke exposure, including cigarette smoke and, in developing countries, the smoke from cooking fires, but it is not clear why individuals with COPD are more susceptible to the influenza virus.

The influenza virus gains entry to lung cells by manipulating receptors on the cell surface. A protein called SOCS5 is present inside these cells and has been suggested as a potential regulator of these receptors. Here, Kedzierski et al. reveal that SOCS5 plays a critical role in protecting lung cells in mice and humans from the virus.

The experiments show that mice lacking the gene that encodes SOCS5 were more susceptible to infection by the influenza virus, had more severe symptoms of disease and increased amounts of virus in their lungs. Further experiments in lung cells collected from human volunteers show that SOCS5 levels increased in both healthy smokers and non-smokers in response to influenza infection. Conversely, SOCS5 levels in lung cells of smokers with COPD remained low after infection. This suggests that SOCS5 might be an important factor in the susceptibility of these patients to influenza.

The next is step is to understand exactly how SOCS5 works, which may make it possible to develop new treatments that boost SOCS5 activity in influenza patients.

existing respiratory conditions such as asthma or chronic obstructive pulmonary disease (COPD) are at increased risk of influenza and suffer more severe clinical symptoms (*Glezen et al., 2000*; *Griffin et al., 2002*).

The global community is constantly on the alert for novel avian influenza strains (*Hansbro et al., 2010*) that might acquire the ability to spill over from birds to humans and against which we have no pre-existing antigenic immunity, thus raising the very real threat of a global pandemic (as occurred during the 1918 Spanish Flu). For example, the virulent H5N1 virus, when transmitted from birds to humans has a 50% mortality rate (*Fauci, 2006*) and since 2013, bird-to-human transmission of the H7N9 virus has resulted in over 600 confirmed cases with a mortality rate of approximately 38%, triggering an emergency response in China (*Hsu et al., 2011*; *Xu et al., 2013*). If such a virus evolved the capacity to transmit efficiently from human-to-human, the effects would be devastating (*Fauci, 2006*).

It is not clear why only some individuals respond to infection with exacerbated inflammatory responses. Similarly, the aetiology surrounding sufferers of chronic respiratory disease and their susceptibility to influenza remains poorly understood (*Hsu et al., 2012b*). We are however, beginning to understand the innate immune sensors that initiate the immune defence to viral pathogens and the ability of the influenza virus to evade and indeed hijack the host defence to facilitate viral entry and replication (*Hsu et al., 2015*).

Encapsulated within the influenza viral envelope are eight RNA segments that encode 11 proteins. Viral entry occurs via attachment of the surface glycoprotein hemagglutinin (HA) to sialic acid residues on the surface of host respiratory epithelial cells. Endocytosis of the viral particles is accompanied by clustering of adjacent receptor tyrosine kinases into lipid rafts. The EGF receptor (EGFR) is one such example, and virus-dependent clustering of the receptor results in activation of EGFR kinase activity and downstream phosphoinositide-3-kinase (PI3K) signaling, which not only facilitates viral entry, but has been shown to suppress interferon regulatory factor (IRF)-induced interferon (IFN) production (*Eierhoff et al., 2010*; *Hsu et al., 2015*; *Ueki et al., 2013*).

Viral RNA generated during replication is detected by host pattern-recognition receptors, specifically by the retinoic-acid induced gene I (RIG-I)-like RNA sensors RIG-I and Melanoma

Differentiation-Associated gene 5 (MDA-5) in the cytoplasm and by Toll-like receptor (TLR)s 3, 7 and 8 in the late endosomes (*Guillot et al., 2005*; *Hsu et al., 2012b*). Activation of these pathways induces production of type I (IFN$\alpha/\beta$) and type III (IFN$\lambda$) interferons and subsequently, the transcription of interferon response genes, which are critical for generating an anti-viral state (*Crotta et al., 2013*; *Seth et al., 2005*). The constitutive production of type I IFNs also contributes to anti-viral immunity (*Hsu et al., 2012a*). Ultimately, TLR and RIG-I signaling pathways converge to induce pro-inflammatory chemokines and cytokines, and recruit a wave of infiltrating immune cells.

Several influenza viral proteins thwart the host response, most notably the multi-functional non-structural 1 (NS1) protein, which interacts with numerous host proteins to inhibit mRNA processing and export, prevent IFN production and alter the intracellular environment (*Marc, 2014*; *Samji, 2009*). For instance, NS1 interacts with the p85 subunit of PI3K to stabilise the kinase complex and promote AKT phosphorylation, enhancing viral entry and inhibiting protective apoptosis (*Ehrhardt and Ludwig, 2009*).

The suppressor of cytokine signaling proteins (CIS, SOCS1-7) are an important family of small intracellular proteins which act to limit the duration of signaling responses. They primarily act as adaptor proteins to recruit an E3 ubiquitin ligase complex and facilitate the ubiquitination of substrates bound to the SOCS-SH2 domain or N-terminal region, marking them for proteasomal degradation (*Linossi and Nicholson, 2012*). While the physiological role and mechanism of action of SOCS1 and particularly SOCS3, are well understood, much less is known about the other family members (SOCS4-7), which in addition to the central phosphotyrosine-binding SH2 domain and C-terminal SOCS box, contain an extended N-terminal region (*Feng et al., 2012*). Although SOCS5 has been suggested to regulate EGF (*Kario et al., 2005*; *Nicholson et al., 2005*) and IL-4 signaling (*Seki et al., 2002*), there is a paucity of data to support a role for SOCS5 as a physiological regulator of these pathways in mammalian cells.

Here we show for the first time that SOCS5 has a unique role in restraining the early phase of influenza A infection in airway epithelial cells. We highlight a novel role for SOCS5 in the regulation of PI3K signaling and demonstrate that SOCS5 primarily protects against viral infection by inhibiting EGFR activity. Given the reduced SOCS5 levels in epithelial cells from COPD patients, this provides a hitherto unknown link that may explain the increased susceptibility of COPD patients to influenza virus infection.

## Results

### *Socs5$^{-/-}$* mice show increased susceptibility to influenza A virus infection

To examine the role of SOCS5 during an infectious challenge, we infected wild-type and *Socs5$^{-/-}$* BALB/c mice with the influenza virus strain A/Puerto Rico/8/32 (H1N1; PR8). Mice lacking *Socs5* lost significantly more body weight following PR8 infection compared to wild-type BALB/c mice (*Figure 1A*). This was evident at day 3 of infection and correlated with a significantly increased viral load in the lung (*Figure 1B*). Interestingly, elevated levels of virus were present in *Socs5$^{-/-}$* lungs from day 1, prior to infiltration of immune cells and suggesting that *Socs5$^{-/-}$* mice had a reduced innate ability to restrain early viral replication. This difference was comparable to the increased levels of virus observed in SirpA-deficient mice which lack natural killer (NK) cells, T and B cells and innate lymphoid cells (ILCs) (*Legrand et al., 2011*), but not as great as that observed in *Socs4$^{-/-}$* mice challenged with PR8 (*Figure 1—figure supplement 1*).

We had previously shown that SOCS4 restrains viral infection via the hematopoietic compartment, most likely through regulating CD8+ T cell function (*Kedzierski et al., 2014*). We therefore investigated the contribution of the hematopoietic compartment to the increased susceptibility to influenza virus observed in the *Socs5$^{-/-}$* mice. Chimeric mice were generated by bone marrow transplantation into irradiated, congenic-recipient mice, which were then challenged with PR8 virus. Transplantation of wild-type bone marrow into *Socs5$^{-/-}$* hosts resulted in greater weight loss and elevated viral titres, when compared to transplantation of *Socs5$^{-/-}$* bone marrow into irradiated wild-type hosts (*Figure 1C,D*). This strongly suggested that the *Socs5$^{-/-}$* defect occurred predominately in non-hematopoietic tissues.

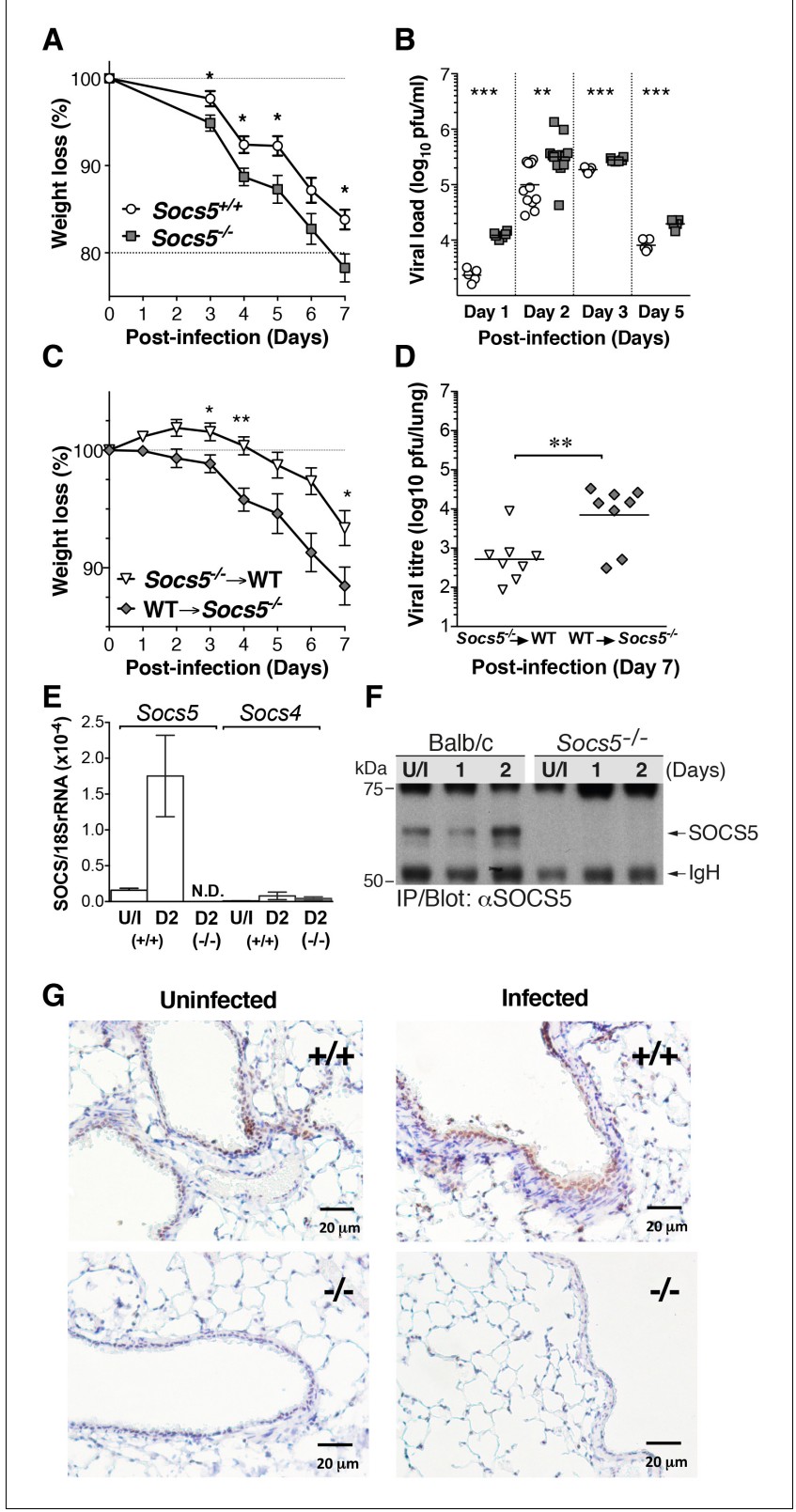

**Figure 1.** *Socs5[−/−]* mice show increased susceptibility to influenza A virus infection. (**A**) Mice were infected i.n. with 25 pfu influenza virus H1N1 PR8 and weight loss monitored for 7 days. Mice that lost more than 20% of their initial body weight were euthanized. Mean ± S.E.M.; n = 9. (**B**) Comparison of viral titres in lung homogenates on days 1, 2, 3 and 5 post-PR8 infection. (**C,D**) Lethally irradiated *Socs5[+/+]* Thy1.1 mice were reconstituted with

*Figure 1 continued on next page*

*Figure 1 continued*

*Socs5*$^{-/-}$ Thy1.2 bone marrow (n = 8) or the reciprocal transplantation was performed (n = 9), following which mice were infected with H1N1 PR8 virus. Weight loss (**C**) was monitored for 7 days, at which time lungs were harvested for (**D**) viral titre estimation. *p<0.05, **<0.005, ***<0.001; mean ± S.E.M. (**E**) Lungs from wild-type and *Socs5*$^{-/-}$ mice were harvested at day (**D**) two post-infection and analyzed for *Socs5* and *Socs4* mRNA expression by Q-PCR. The results were normalized to 18SrRNA levels. Mean ± S.E.M.; N.D.=Not detected; n = 3 uninfected (U/I) wild-type mice, n = 6 infected wild-type mice, n = 6 infected *Socs5*$^{-/-}$ mice. (**F**) Lungs were lysed at day 1 and 2 post-infection and SOCS5 protein levels analyzed by immunoprecipitation and immunoblotting. (**G**) Lung immunohistochemistry showing SOCS5 expression (brown staining) in wild-type (+/+) airway epithelium at day three post-infection. The following figure supplement is available for *Figure 1*.

The following figure supplement is available for figure 1:

**Figure supplement 1.** Comparison of viral titres.

## *Socs5* is expressed in airway epithelial cells and is upregulated in response to influenza virus infection

*Socs5* mRNA was expressed in uninfected mouse lungs and was significantly upregulated at day two post-infection; by comparison, *Socs4* was expressed at very low levels even during infection (*Figure 1E*). These data were confirmed at the protein level by immunoprecipitation and immunoblotting with anti-SOCS5 antibodies, which detected a prominent band migrating at ~67 kDa in wild-type, but not *Socs5*$^{-/-}$ lungs (*Figure 1F*). Immunohistochemistry demonstrated specific staining in wild-type lungs, which was increased during infection and was predominately localized to the airway epithelial cells lining the bronchioles (*Figure 1G*).

## Increased influenza severity in the *Socs5*$^{-/-}$ mice is associated with increased inflammation and neutrophil infiltration

Pro-inflammatory cytokines and chemokines were elevated in the bronchoalveolar lavage (BAL) from *Socs5*$^{-/-}$ mice, day two post-infection. In particular, the cytokines interleukin (IL)-6 and G-CSF, and the chemokines KC, MCP-1 and MIP-1$\beta$ were elevated compared to controls (*Figure 2A*). In contrast, type I and type III IFNs were not increased in *Socs5*$^{-/-}$ lung homogenates, whilst the levels of IFN$\alpha$, $\beta$ and $\lambda$ appeared to be modestly decreased at day one post-infection (*Figure 2B*).

There was also an increase in the total number of cells infiltrating into the airways (*Figure 2C*). This was accounted for by an increase in neutrophils and is consistent with the elevated cytokine/chemokine levels, in particular the known roles of IL−6 and G-CSF in neutrophil activation and survival, and of KC and MCP-1 in neutrophil recruitment (*Soehnlein and Lindbom, 2010*). There were no differences observed in infiltrating monocytic cells, T or B cells (*Figure 2C* and *Figure 2—figure supplement 1*). At day two post-infection, these changes were apparent at a global level in *Socs5*$^{-/-}$ lungs, with quantitative proteomic analysis showing increased expression of neutrophil proteins and neutrophil chemotactic proteins, in addition to detection of viral NS1, HA and NP proteins (*Figure 2D,E* and *Table 1*). A total of 1907 unique mouse proteins were identified, with 23 differentially regulated in *Socs5*$^{-/-}$ lungs. Interestingly, a number of histones were also upregulated in *Socs5*$^{-/-}$ lungs. Together with increased Hmgb2 and various neutrophil effector proteins, this signature is strongly reminiscent of neutrophil extracellular traps (NETs) (*Khandpur et al., 2013*; *Urban et al., 2009*), a mechanism whereby dying neutrophils extrude DNA 'nets' coated with nuclear and granular proteins, to trap and kill the invading microorganisms (*Rohrbach et al., 2012*) (*Figure 2D,E* and *Table 1*). This was further supported by the increased amount of extracellular DNA found in *Socs5*$^{-/-}$ lungs (*Figure 2—figure supplement 2*) and by the detection of citrullinated modifications on the differentially expressed proteins (lactotransferrin, neutrophil elastase) (*Table 1*) (*Wang et al., 2009*).

Collectively, these data suggest that the heightened susceptibility to influenza observed in *Socs5*$^{-/-}$ mice results from a defect in the lung epithelium, with the increased viral titre in the *Socs5*$^{-/-}$ lungs initiating an inflammatory cascade that is driven primarily by neutrophil infiltration (*Figure 2F*).

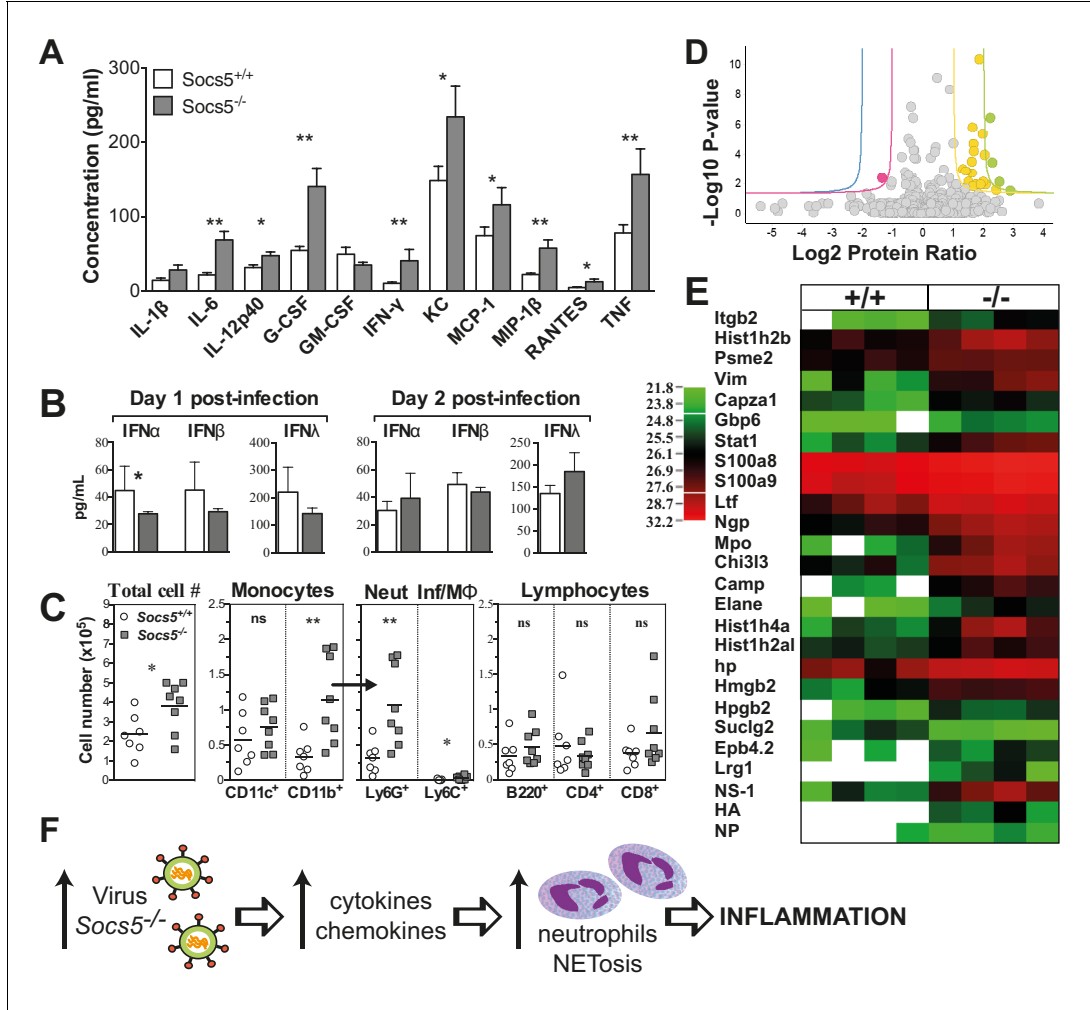

**Figure 2.** *Socs5^−/−* mice have an exaggerated inflammatory response in the lungs to influenza A virus infection. (**A**) Cytokine and chemokine levels were analyzed by Bioplex in bronchoalveolar lavage (BAL) fluid recovered from lungs at day two post-infection with 25 pfu influenza virus H1N1 PR8. *p<0.05, **<0.005; Mean ± S.E.M.; n = 7 for *Socs5^+/+^*, n = 8 for *Socs5^−/−^*. (**B**) Expression of type I and type III interferon (IFN) in the lungs at day 1 and day two post-infection was measured by ELISA. *p<0.05; Mean ± S.D.; day 1: n = 5 for *Socs5^+/+^*, n = 6 for *Socs5^−/−^*; day 2: n = 8. (**C**) Flow cytometric analysis was performed on cells in BAL. Cells were gated on CD11b^+^ and subdivided into CD11c^+^ APC (dendritic cells and alveolar macrophages), Ly6G^+^ neutrophils and Ly6C^+^ inflammatory monocytes. Gated lymphocytes were subdivided into B220^+^ B cells and CD4^+^ and CD8^+^ T cells. *p<0.05, **<0.005, ns = not significant; Individual and mean values; n = 7 for *Socs5^+/+^*, n = 8 for *Socs5^−/−^*. (**D,E**) Mice were infected and lungs lysed at day two post-infection for label-free quantification of global protein expression. Volcano plot (**D**) shows the Log2 protein ratios following the quantitative pipeline analysis (*Socs5^+/+^* vs *Socs5^−/−^*). The red and yellow lines represent a 2-fold change in protein expression (log2 ratio of 1), while blue and green lines represent a 4-fold change (log2 ratio of 2); dots are coloured accordingly and represent individual proteins. Proteins with a -log10 p-value of 1.3 or greater (corresponding to a p-value of ≤0.05) were deemed differentially abundant. (**E**) Heat map displaying Log2-transformed summed peptide intensities (non-imputed) for proteins with significantly differential expression in (**D**). Data from individual biological replicates are shown (n = 4). Green to red indicates increasing expression levels. (**F**) Schematic showing the relationship between increasing viral load and inflammation.

The following figure supplements are available for figure 2:

**Figure supplement 1.** Gating strategy for flow cytometric analysis.

**Figure supplement 2.** Increased NETosis in *Socs5^−/−^* lungs during influenza virus infection.

**Table 1.** Quantitative proteomic analysis showing differentially expressed proteins in lung lysates at day two post-infection.

| Accession number | Protein names | Gene name | Log2 protein ratio (−/−):(+/+) | Protein P-Value (−/−): (+/+) | #unique peptides (−/−) | #unique peptides (+/+) | |
|---|---|---|---|---|---|---|---|
| Q3UP42 | Protein S100-A9 | S100a9 | 2.90 | 2.94E-02 | 2 | 2 | NET-associated protein |
| P62806 | Histone H4 | Hist1h4a | 2.53 | 7.82E-03 | 4 | 3 | NET-associated protein |
| Q3UP87 | Neutrophil elastase* | Elane | 2.43 | 2.66E-02 | 2 | 1 | NET-associated protein |
| Q53 × 15 | Protein S100-A8 | S100a8 | 2.30 | 3.38E-04 | 3 | 3 | NET-associated protein |
| Q91XL1 | Leucine-rich alpha-2-glycoprotein | Lrg1 | 2.07 | 1.14E-02 | 3 | 0 | Neutrophil differentiation |
| Q61646 | Haptoglobin | Hp | 2.05 | 1.03E-04 | 13 | 9 | Produced by neutrophils |
| Q542I8 | Integrin beta 2 (Mac-1) | Itgb2 | 2.00 | 7.85E-03 | 6 | 1 | Induces Net formation |
| Q7TMS4 | Myeloperoxidase | Mpo | 1.97 | 4.18E-06 | 14 | 2 | NET-associated protein |
| P51437 | Cathelin-related antimicrobial peptide | Camp | 1.90 | 1.10E-02 | 3 | 1 | Highly expressed in neutrophils |
| Q4FJR3 | Lactotransferrin* | Ltf | 1.87 | 4.43E-11 | 26 | 13 | NET-associated protein |
| O08692 | Neutrophilic granule protein | Ngp | 1.76 | 1.10E-02 | 6 | 4 | In neutrophil granules |
| Q99K94 | Signal transducer and activator of transcription 1 | Stat1 | 1.68 | 5.80E-05 | 11 | 4 | Low in neutrophils – could be upregulated in response to IFN or EGFR? |
| P20152 | Vimentin | Vim | 1.67 | 1.99E-05 | 14 | 3 | NET-associated protein |
| Q3UV87 | Chitinase-3-like protein 3 (YM1) | Chi3l3 | 1.65 | 1.53E-06 | 9 | 6 | Produced by neutrophils |
| A0JLV3 | Histone H2B | Hist1h2bj | 1.32 | 9.49E-04 | 3 | 3 | NET-associated protein |
| G3 × 9V0 | Proteasome activator complex subunit 2 | Psme2 | 1.63 | 1.36E-02 | 7 | 5 | |
| Q5RKN9 | F-actin-capping protein subunit alpha-1 | Capza1 | 1.65 | 1.48E-02 | 4 | 2 | Neutrophil protein which inhibits actin polymerisation |
| Q8VCC1 | 15-hydroxyprostaglandin dehydrogenase [NAD(+)] | Hpgd | 1.63 | 1.87E-03 | 3 | 1 | Not in neutrophils? |
| Q3UAZ7 | High mobility group protein B2 | Hmgb2 | 1.62 | 3.03E-04 | 4 | 2 | Hmgb1: component of neutrophil NET filaments/induces NET formation |
| Q8BH61 | Erythrocyte membrane protein band 4.2 | Epb4.2 | 1.51 | 7.57E-03 | 5 | 1 | Neutrophil protein |
| F8WIX8 | Histone H2A | Hist1h2al | 1.44 | 1.73E-02 | 2 | 2 | NET-associated protein |
| Q6PEN2 | Protein Gbp6 | Gbp6 | 1.41 | 1.36E-03 | 3 | 1 | Not in neutrophils? |
| Q9Z2I8-2 | Succinyl-CoA ligase subunit beta | Suclg2 | −1.36 | 4.19E-03 | 1 | 3 | |

* indicates the detection of citrullinated peptides.

Note: Given that trypsin cleaves at arginine and lysine residues and citrullination is a modification of arginine, it is likely that citrullination impacts on the efficiency of the tryptic digest. Hence, we may be underestimating the number of citrillinated proteins present.

### Socs5<sup>−/−</sup> primary mouse airway epithelial cells (mAEC) have increased intrinsic susceptibility to influenza virus infection

To determine whether the enhanced susceptibility to infection was intrinsic to the pulmonary epithelial cells, lungs were harvested from wild-type and $Socs5^{-/-}$ mice and purified mAECs cultured for 7 days prior to infection with influenza virus PR8 strain (**Figure 3A**). At low levels of virus (MOI 1.25 and 2.5), there was a higher percentage of infected $Socs5^{-/-}$ mAECs in comparison to wild-type cells (**Figure 3B**), suggesting that $Socs5^{-/-}$ mAECs may be more permissive for viral infection. However, this difference was lost once the initial inoculum was increased (MOI 5 and 10; >74% infected cells). When a low MOI was used in the presence of trypsin (to ensure multiple rounds of replication

and release from the cells), *Socs5*$^{-/-}$ mAECs showed elevated viral titres 24 and 48 h post-infection, confirming the reduced capacity of *Socs5*$^{-/-}$ mAECs to restrain viral replication (*Figure 3C*). Consistent with our in vivo results, there was increased cytokine and chemokine production, with elevated IL-6, G-CSF, MCP-1, MIP-1α and RANTES at 24 h post-infection (MOI 5). Of these, G-CSF and MCP-1 showed the greatest differential expression (*Figure 3D*).

These results provide evidence for an intrinsic epithelial cell defect, which manifests in increased levels of virus and is likely to be, at least in part, responsible for the phenotype observed in *Socs5*$^{-/-}$ mice (*Figure 1*).

## SOCS5 regulates EGF-PI3K signaling

To investigate which pathways might be perturbed in the absence of *Socs5*, whole lung lysates were analyzed by immunoblotting for the activation of various anti- and pro-viral signaling intermediates. Given that exogenous expression of SOCS5 has been shown to downregulate EGFR levels (*Nicholson et al., 2005*), we included analysis of EGFR expression. Lungs from uninfected *Socs5*$^{-/-}$ mice showed increased expression of the p110α catalytic subunit of PI3K, and apart from slightly reduced levels of phosphorylated (p)AKT and PI3K p85, other signaling proteins remaining unchanged (*Figure 4A,B*). At day one post-infection, the expression of EGFR and PI3K p85 and p110α subunits were increased in *Socs5*$^{-/-}$ lungs. Further, there was also an increase in pAKT and pSTAT3. In contrast, the levels of pMAPK and total RIG-I were comparable in wild-type and *Socs5*$^{-/-}$ lungs (*Figure 4A,B*).

To determine whether SOCS5 existed in the same protein complexes as PI3K, Flag-tagged SOCS5 was expressed in 293T cells, immunoprecipitated and mass spectrometry used to interrogate protein complex composition (*Figure 5A* and *Table 2*). SOCS box components (Elongins B and C, Rbx2 and Cullin5) were specifically enriched in SOCS5 immunoprecipitates, but not from cells transfected with vector alone. Similarly, the PI3K subunits p85 (PIK3R1), p85β (PIK3R2) and p110β (PIK3CB) were enriched in SOCS5 complexes (*Figure 5A* and *Table 2*). We confirmed these results by immunoblotting SOCS5 complexes. The SOCS5:PI3K (p85; p110α/β) complex was enriched in cells pre-treated with vanadyl hydroperoxide (pervanadate) (to block phosphatase activity), indicating that SOCS5 interaction with the PI3K complex was dependent on phosphorylation of one of the components (*Figure 5B*). Further, mutation of the SOCS5-SH2 domain (R406K) or deletion of the N-terminal 349 residues (ΔNT) reduced SOCS5 interaction with the PI3K complex, whereas mutation of the SOCS box (L484P, C488F) had no effect on binding. Importantly, we utilised the lung epithelial A549 cell line to confirm, by co-immunoprecipitation, that endogenous SOCS5 and PI3K co-existed in a protein complex (*Figure 5D*).

This suggests that the EGFR and PI3K signaling complexes might be direct targets for SOCS5 regulation and that recruitment of the PI3K complex is mediated, at least in part, via a canonical SOCS5-SH2:phosphotyrosine interaction. Notably, the difference in PI3K p110α expression in lungs was detected prior to infection, suggesting that exaggerated activation of this pathway might underlie the increased viral susceptibility in the *Socs5*$^{-/-}$ mice. In contrast, the changes in EGFR levels and AKT/STAT3 phosphorylation appeared to be influenza virus-induced effects, albeit dependent on SOCS5.

## SOCS5 expression in primary human airway epithelial cells correlates with increased susceptibility to influenza virus infection

It was important to ascertain the role of SOCS5 in human disease and understand the consequences of changes in SOCS5 expression. This was particularly relevant given the previous finding that PI3K activity was elevated in AECs from COPD patients (*Hsu et al., 2015*).

Primary human airway epithelial cells (hAECs) were obtained by endobronchial brushing from healthy individuals, smokers with no evidence of lung disease (smokers) and ex-smokers suffering from COPD, and examined by Q-PCR for *Socs5* expression following infection with various human (H3N2, H1N1; MOI 5) and avian (H11N9; MOI 5) influenza strains. In cells from healthy individuals, SOCS5 was elevated in response to influenza virus infection with all three strains. However, in cells obtained from individuals with COPD, *Socs5* barely increased above basal levels and was significantly lower than in hAECs from healthy individuals (non-smokers or smokers) (*Figure 6A*). In

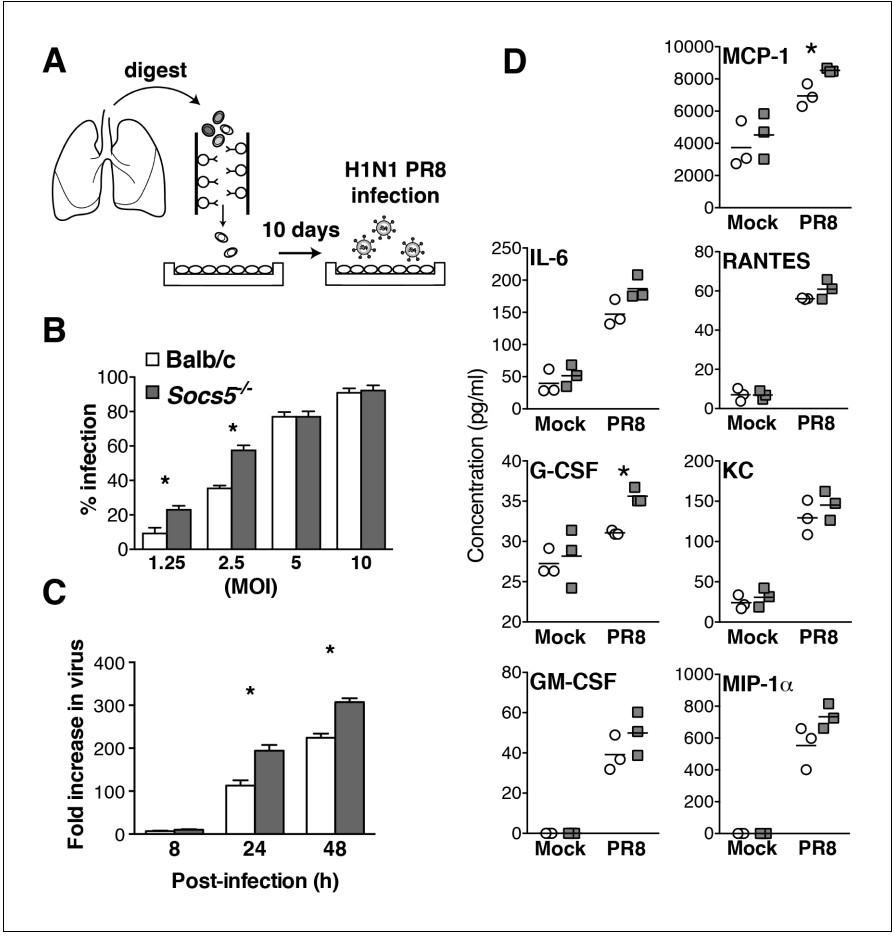

**Figure 3.** *Socs5*$^{-/-}$ airway epithelial cells have increased intrinsic susceptibility to influenza virus infection. (**A**) Primary mouse airway epithelial cells (mAEC) from wild-type and *Socs5*$^{-/-}$ mice were purified and cultured in vitro for 7 days prior to infection with influenza virus H1N1 PR8. (**B**) 8 h post-infection cell monolayers were fixed and stained by immunofluorescence for detection of viral nuclear protein. Results are shown as the mean % of infected cells from four technical replicates and are representative of 2 experiments. (**C**) mAEC were infected with PR8 (MOI 1) and then incubated in the presence of trypsin for 2, 8, 24 and 48 h. Culture supernatants were analyzed for infectious virus by plaque assay. Results are shown as the fold increase over the level of virus present at 2 h post-infection. Mean ± S.E.M. are shown for three technical replicates and are representative of 2 experiments. (**D**) mAECs were infected (MOI 5) and cytokine levels in culture supernatants measured by Bioplex at 24 h post-infection. Individual and mean values are shown for technical replicates derived from purified cells pooled from five mice and are representative of 2 experiments. *p<0.05.

contrast to infection with H3N2, H1N1 and H11N9, infection with the highly pathogenic avian virus H5N1 dramatically suppressed *Socs5* expression relative to the media control (*Figure 6B*).

We next sought to test the hypothesis that SOCS5 might be regulating viral infection via negative regulation of EGFR and/or PI3K signaling. As an initial step, we examined levels of SOCS5 protein, phosphorylated EGFR, PI3K p110α/p85 and phosphorylated AKT in hAECs from healthy individuals, and COPD patients (*Figure 6C–F*, and *Figure 6—figure supplement 1*). SOCS5 protein was induced in healthy cells 24 h post-H1N1 infection (*Figure 6C,D*) and was consistently reduced in cells from COPD patients, both at a basal level and with limited induction in response to virus (*Figure 6C–F* and *Figure 6—figure supplement 1A,B*). SOCS5 expression inversely correlated with p110α levels and virus-induced EGFR phosphorylation, AKT phosphorylation and viral protein, which was elevated in COPD hAECs as indicated by detection of intracellular viral HA protein (*Figure 6—figure supplement 1A,B*).

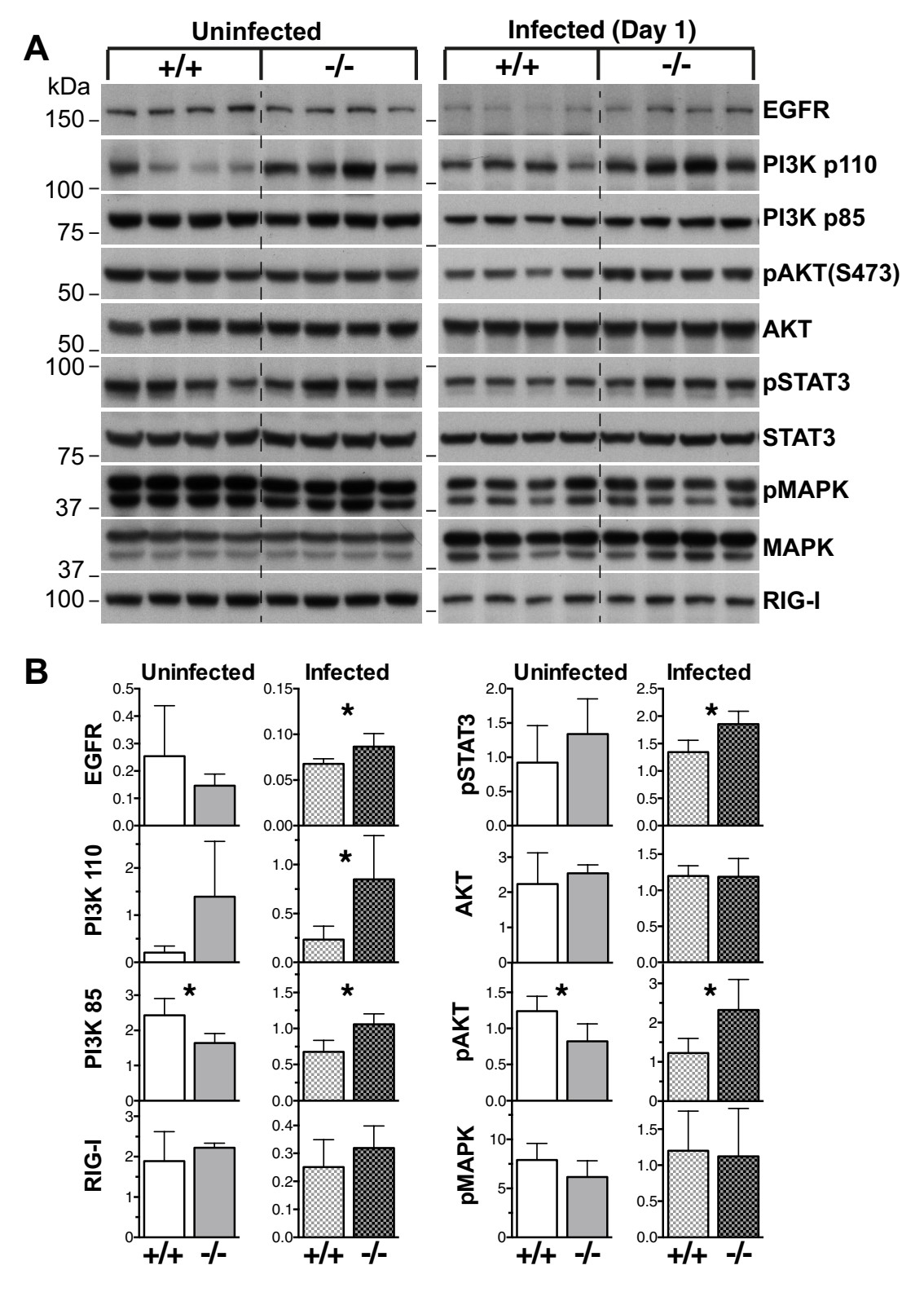

**Figure 4.** EGF-PI3K signaling is enhanced in *Socs5*$^{-/-}$ lungs. (**A**) Whole lungs were harvested from uninfected wild-type (+/+) and *Socs5*$^{-/-}$ (−/−) mice or from mice at 24 h post-infection with 50 pfu influenza virus H1N1 PR8. Following tissue lysis 100 µg protein from each mouse was analyzed by immunoblotting with antibodies to the indicated proteins. p: phosphorylated. Note that as they are analyzed on separate gels, uninfected samples

*Figure 4 continued on next page*

*Figure 4 continued*

cannot be compared to infected samples. (B) Densitometric values derived from (A) were analyzed as follows: Data from uninfected mice were normalized to MAPK values, whilst data from infected mice were normalized to STAT3. *p<0.05, **p<0.005; Mean ± S.D.; n = 4.

To determine whether SOCS5 expression -was functionally linked to pathogenicity, hAECs were cultured from healthy individuals or COPD patients and SOCS5 either reduced using siRNA (SOCS5-si) or increased by transient transfection (SOCS5) (*Figure 6C–F* and *Figure 6—figure supplements 1* and *2*). Depletion of SOCS5 resulted in enhanced levels of the PI3K p110α catalytic subunit in both healthy and COPD hAECs and this occurred independently of virus. Depletion of SOCS5 also resulted in increased viral-induction of EGFR phosphorylation, and the changes in EGFR and PI3K again correlated with increased viral HA levels 2 h post-infection. Phosphorylation of AKT was also enhanced in some experiments (*Figure 6—figure supplements 1C*, *2* and *3*). Conversely, forced SOCS5 expression resulted in reduced p110α and EGFR levels and reduced EGFR phosphorylation.

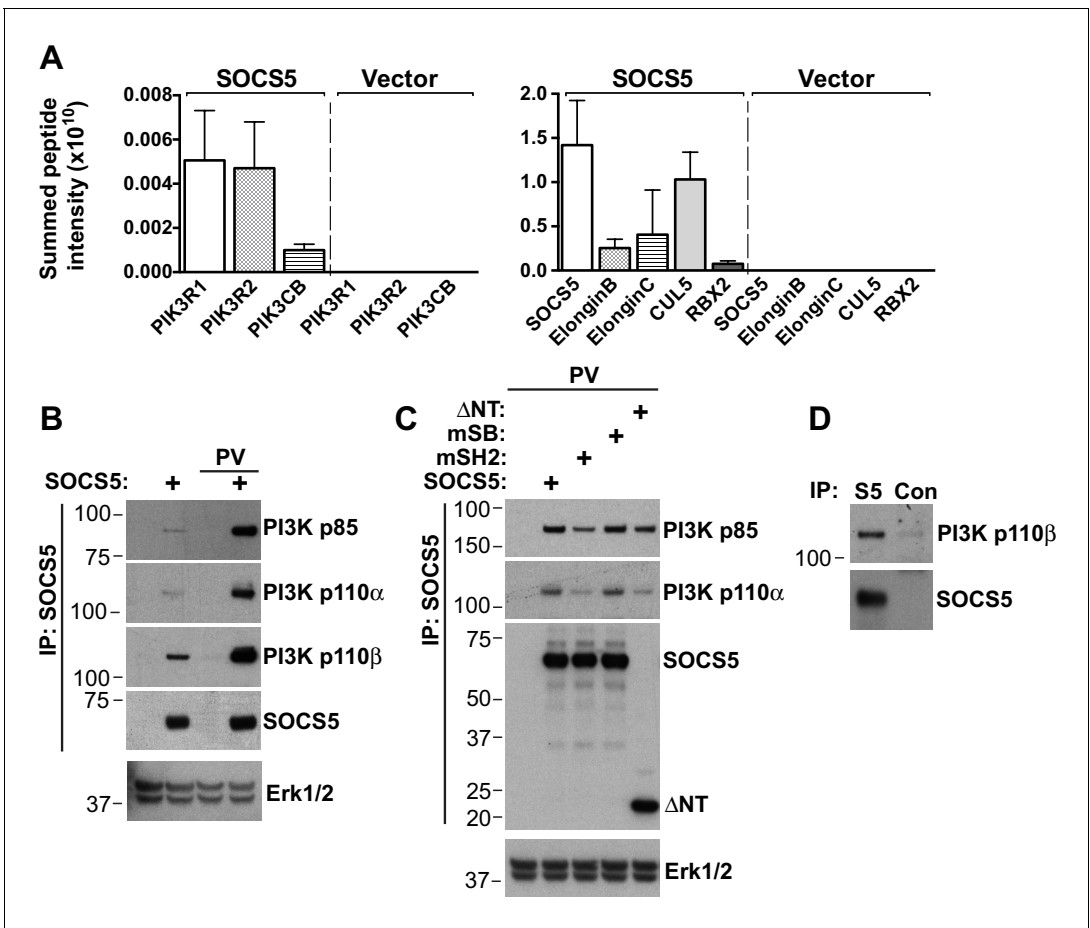

**Figure 5.** SOCS5 interacts with a PI3K complex. (A–C) 293T cells were transiently transfected with constructs expressing Flag-tagged SOCS5, SOCS5 mutants or vector alone, lysed and SOCS5-containing complexes immunoprecipitated (IP) using anti-Flag antibodies. (A) Protein complexes were analyzed by tryptic digest and mass spectrometry. Mean ± S.D. of summed peptide intensities from three replicates. The PI3K components, and SOCS5 and SOCS box components, were specifically enriched relative to immunoprecipitates from cells transfected with vector alone. (B,C) SOCS5-containing protein complexes were analyzed by immunoblotting with the indicated antibodies. Lysates were probed with antibodies to Erk1/2 to show equal protein input (lower panels). PV indicates pre-treatment of cells with pervanadate to inhibit phosphatase activity. mSH2: SOCS5-R406K; mSB: SOCS5-L484P, C488F; ΔNT: residues 350–536. (D) A549 cells were pre-treated with the proteasomal inhibitor MG132 (6 h) and pervandate (20 min) prior to lysis. Proteins were immunoprecipitated with either isotype control antibodies (Con) or SOCS5-specific antibodies (S5) and analyzed by immunoblotting with the indicated antibodies.

**Table 2.** Proteomic analysis of SOCS5 immunoprecipitates.

| Accession number | Protein names | Gene names | Summed peptide intensity | | | | | | #unique peptides | | | | | |
|---|---|---|---|---|---|---|---|---|---|---|---|---|---|---|
| | | | SOCS5* | | | Vector | | | SOCS5 | | | Vector | | |
| P27986 | PI3K p85 $\beta$ | PIK3R2 | 5.63E + 07 | 2.31E + 07 | 6.18E + 07 | 0 | 0 | 0 | 6 | 2 | 7 | 0 | 0 | 0 |
| O00459 | PI3K p85 $\alpha$ | PIK3R1 | 5.57E + 07 | 2.60E + 07 | 7.02E + 07 | 0 | 0 | 0 | 7 | 2 | 6 | 0 | 0 | 0 |
| P42338 | PI3K p110 $\beta$ | PIK3CB | 8.07E + 06 | 1.00E + 00 | 1.18E + 07 | 0 | 0 | 0 | 3 | 0 | 4 | 0 | 0 | 0 |
| O54928 | SOCS5 | SOCS5 | 1.06E + 10 | 1.20E + 10 | 2.00E + 10 | 0 | 0 | 0 | 32 | 30 | 37 | 0 | 0 | 0 |
| Q93034 | Cullin-5 | CUL5 | 1.08E + 10 | 7.01E + 09 | 1.31E + 10 | 0 | 0 | 0 | 54 | 46 | 51 | 0 | 0 | 0 |
| Q9UBF6 | RBX2 | RNF7 | 7.23E + 08 | 4.36E + 08 | 1.10E + 09 | 0 | 0 | 0 | 4 | 3 | 4 | 0 | 0 | 0 |
| Q15369 | EloninB | TCEB1 | 1.41E + 09 | 9.89E + 08 | 8.69E + 08 | 0 | 0 | 4.78E + 06 | 5 | 5 | 5 | 0 | 0 | 1 |
| Q15370 | ElonginC | TCEB2 | 2.92E + 09 | 1.40E + 09 | 3.30E + 09 | 0 | 0 | 0 | 10 | 7 | 8 | 0 | 0 | 0 |

\* Data are shown from replicate samples.

Consistent with reduced PI3K p110α, SOCS5 also suppressed AKT phosphorylation in both healthy and COPD hAECs (*Figure 6F* and *Figure 6—figure supplements 1D* and *2*). Viral HA levels were also significantly reduced (*Figure 6F* and *Figure 6—figure supplements 1D* and *2*).

The changes in HA levels (within 2 h; *Figure 6E* and *Figure 6—figure supplements 1C*, *2* and *3*) are occurring prior to viral replication, suggesting that SOCS5 impacts on very early events in the viral life cycle. To explore this further, the experiment was repeated in the presence of PI3K (Wortmannin) and EGFR (Erlotinib) inhibitors. Both inhibitors normalized the *Socs5-si* difference (*Figure 6—figure supplement 3*), suggesting that the increase in viral HA protein at this early timepoint derives from EGFR-mediated PI3K activity.

## EGF receptor kinase activity mediates the enhanced susceptibility to influenza infection in the absence of *Socs5*

We then investigated whether increases in EGFR or PI3K signaling caused the increased susceptibility to influenza infection in the absence of *Socs5*. Wild-type and *Socs5*$^{-/-}$ mice were treated with either vehicle control (captisol), PI3K inhibitor (BKM-120) or EGFR inhibitor (Erlotinib) followed by infection with H1N1 PR8 virus. *Socs5*$^{-/-}$ mice again exhibited elevated levels of virus in the lungs at 24 h post-inoculation, compared to controls. Treatment with the PI3K inhibitor significantly reduced the amount of virus in both wild-type and *Socs5*$^{-/-}$ mice, with levels in *Socs5*$^{-/-}$ mice remaining higher than that of wild-type mice. In contrast, treatment with the EGFR inhibitor Erlotinib selectively reduced levels of virus in *Socs5*$^{-/-}$ mice, to the extent that they dropped below levels in wild-type mice (*Figure 7A*).

Remarkably, the key observations were reproduced in both primary mouse and human epithelial cell cultures following infection with H1N1 virus. PI3K inhibition (BKM-120 or Wortmannin) reduced viral titres in primary mouse and human epithelial cells, with SOCS5-deficient cells retaining a significantly higher level of infection compared to wild-type cells (*Figures 7B and C*). In contrast, inhibition of EGFR activity (using either AG1478 or Erlotinib) reduced and normalized viral titres in wild-type and *Socs5*$^{-/-}$ cells (*Figures 7B and C*).

Several key conclusions can be drawn from the inhibitor studies. Firstly, consistent with published data, PI3K signaling is important for viral fitness. Secondly, despite the validity of PI3K as a potential SOCS5 target, the increased susceptibility of *Socs5*$^{-/-}$ mice to influenza is due to altered EGFR signaling. In this context the requirement for EGFR kinase activity predominates over EGFR-mediated PI3K activity.

In summary, these data encapsulate several important findings. SOCS5 expression is upregulated in response to influenza A virus infection in hAECs, suggesting that it has an important role in modulating innate host defense. SOCS5 levels are reduced in hAECs from COPD patients and a highly pathogenic avian virus (H5N1) actively reduced SOCS5 expression, thus linking SOCS5 to influenza pathogenesis. Finally, we have shown in vivo and in primary human cells, that SOCS5 acts to

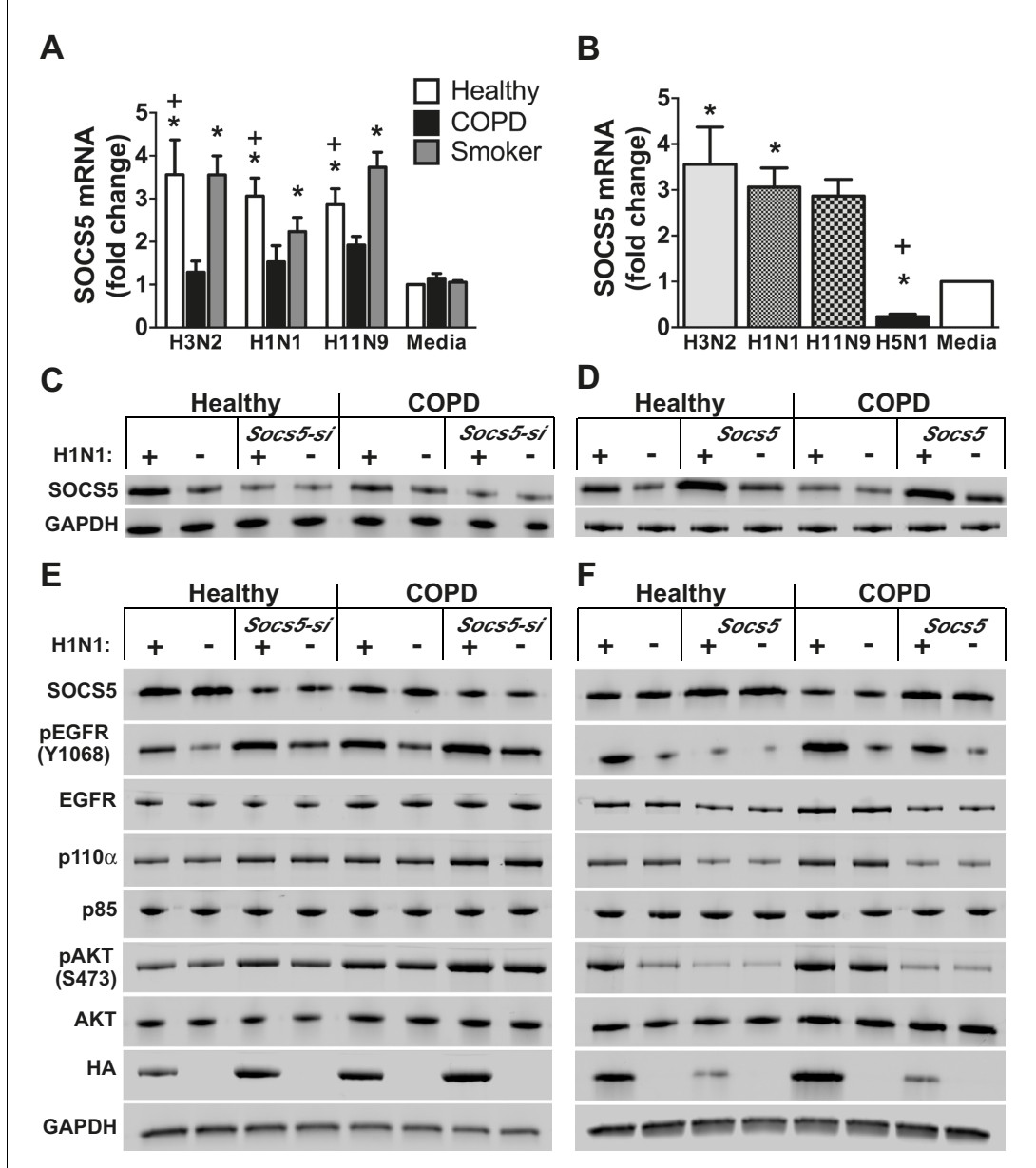

**Figure 6.** *Socs5* expression in primary human airway epithelial cells regulates EGFR-PI3K signaling and restrains influenza virus infection. Primary human airway epithelial cells (hAECs) from healthy individuals, smokers with COPD and smokers without lung disease, were cultured as described and infected with H3N2, H1N1 and H11N9 influenza virus strains (MOI 5). (A) 24 h post-infection *Socs5* mRNA was measured by Q-PCR and is shown as the fold-change from a media only control. Mean ± S.E.M.; n = 3. (B) Data for healthy controls shown in (A) is plotted against the expression of *Socs5* mRNA following infection with H5N1 (MOI 0.005). *p<0.05 Infection vs Media, +p<0.05 healthy vs COPD and H5N1 vs other subtypes/strains; Mean ± S.E.M.; n = 3. (C–F) hAECs from healthy individuals and COPD patients were cultured and SOCS5 depleted using siRNA (C,E). Alternatively, SOCS5 expression was increased in hAECs by transfection with SOCS5 cDNA (SOCS5 vector) (D,F) prior to infection with H1N1 (MOI 5). Cells were lysed at 24 h (C,D) or 2 h (E,F) post-infection and analyzed by immunoblotting with the indicated antibodies.

The following figure supplements are available for figure 6:

**Figure supplement 1.** *Socs5* expression in human primary airway epithelial cells regulates EGFR-PI3K signalling and restrains influenza infection.

**Figure supplement 2.** *Socs5* expression in human primary airway epithelial cells regulates EGFR-PI3K signalling and restrains influenza infection.

**Figure supplement 3.** *Socs5* expression in human primary airway epithelial cells regulates early events in influenza infection.

*Figure 6 continued on next page*

*Figure 6 continued*

**Figure supplement 4.** PI3K p110α is not targeted by SOCS5 for proteasomal degradation.

**Figure supplement 5.** SOCS5 does not ubiquitinate PI3K p110α in vitro.

negatively regulate EGFR and PI3K signaling and while both are important positive mediators of influenza virus infection, it is SOCS5 restriction of EGFR activity, which limits viral infection in lung epithelium.

## Discussion

Our data position SOCS5 as a pivotal regulator of influenza, a major global disease. Mortality related to infection with highly pathogenic influenza strains is associated with the production of high levels of inflammatory cytokines in the airways (in particular IL-6): the 'cytokine storm' (*de Jong et al., 2006*). COPD is a progressive inflammatory disease of the airways that results from smoke exposure (cigarette smoking and biomass smoke) and individuals suffering from COPD are at risk of severe complications following infection with influenza and other respiratory viruses. We suggest that SOCS5 acts as a central switch point, with changes in SOCS5 levels and/or activity impacting disease outcome in individuals such as those with COPD, who exhibit increased susceptibility to influenza.

The $Socs5^{-/-}$ mice present a novel model of exacerbated human influenza, displaying increased viral load, exaggerated cytokine production in the early phase of the infection and increased neutrophil trafficking into the airways (*Figures 1* and *2*). Although total depletion of neutrophils results in worsened disease (*Tate et al., 2011*), their role is more complex. Neutrophils are strongly associated with the inflammatory pathology observed in severe infections (*Brandes et al., 2013*; *Narasaraju et al., 2011*; *Perrone et al., 2008*), and the process of NETosis has recently been shown to induce inflammatory responses (*McIlroy et al., 2014*). Our results suggest that in $Socs5^{-/-}$ mice, amplification of the inflammatory cascade occurs as the neutrophils die via NETosis, releasing pro-inflammatory molecules such as HMBG1 and 2.

Influenza virus-induced EGFR, PI3K and AKT activity have each separately been shown to increase viral entry and replication (*Ehrhardt and Ludwig, 2009*; *Eierhoff et al., 2010*; *Hsu et al., 2015*). Here we link all three to the susceptibility and pathology associated with COPD patients and identify SOCS5 as a key negative regulator of these pathways. While the individual differences observed in AECs may appear modest, the defects in EGFR, PI3K and AKT need to be considered in combination and importantly, in the context of the entire lung. If these small differences occur across a large part of the whole airways in COPD, then the airways will become more susceptible to influenza, as will the patient. There is a fine balance between resistance and susceptibility to infection and these changes could alter the balance towards infection, especially when combined with other immune defects in COPD. Another example is the modest difference seen in IFN signaling in SOCS1-deficient cells (*Brysha et al., 2001*; *Wormald et al., 2006*). While these differences appear small, the consequences in vivo are devastating, SOCS1-deficient mice die of an IFN-driven inflammatory disease shortly after birth (*Alexander et al., 1999*) and similarly $Socs1^{-/-}$ $Ifng^{-/-}$ mice die rapidly when injected with IFNγ (*Brysha et al., 2001*).

Influenza virus binding to sialic acids on the epithelial cell surface, results in clustering of lipid rafts and crosslinking of surface receptors such as c-Met, PDGF and EGFR (*Eierhoff et al., 2010*). The Class I PI3Ks consist of a heterodimer between a regulatory (p85) and a catalytic (p110) subunit of which there are multiple variants (*Cantley, 2002*). Phosphorylated EGFR recruits p85 PI3K to the receptor and activation of PI3K results in phosphorylation of phospholipids at the cell membrane, facilitating endocytosis of the virus, presumably through reorganisation of the actin cytoskeleton (*Ehrhardt et al., 2006*). In addition to this important role for PI3K signaling at the early stages of infection, at later stages direct binding of the viral NS1 protein to PI3K enhances its stability and activation, reducing cell death and maintaining a replicative niche for the virus (*Hale et al., 2006*). EGFR signaling can also suppress the host anti-viral response, inhibiting RIG-I-driven type I interferon production by an as yet, unknown mechanism (*Ueki et al., 2013*).

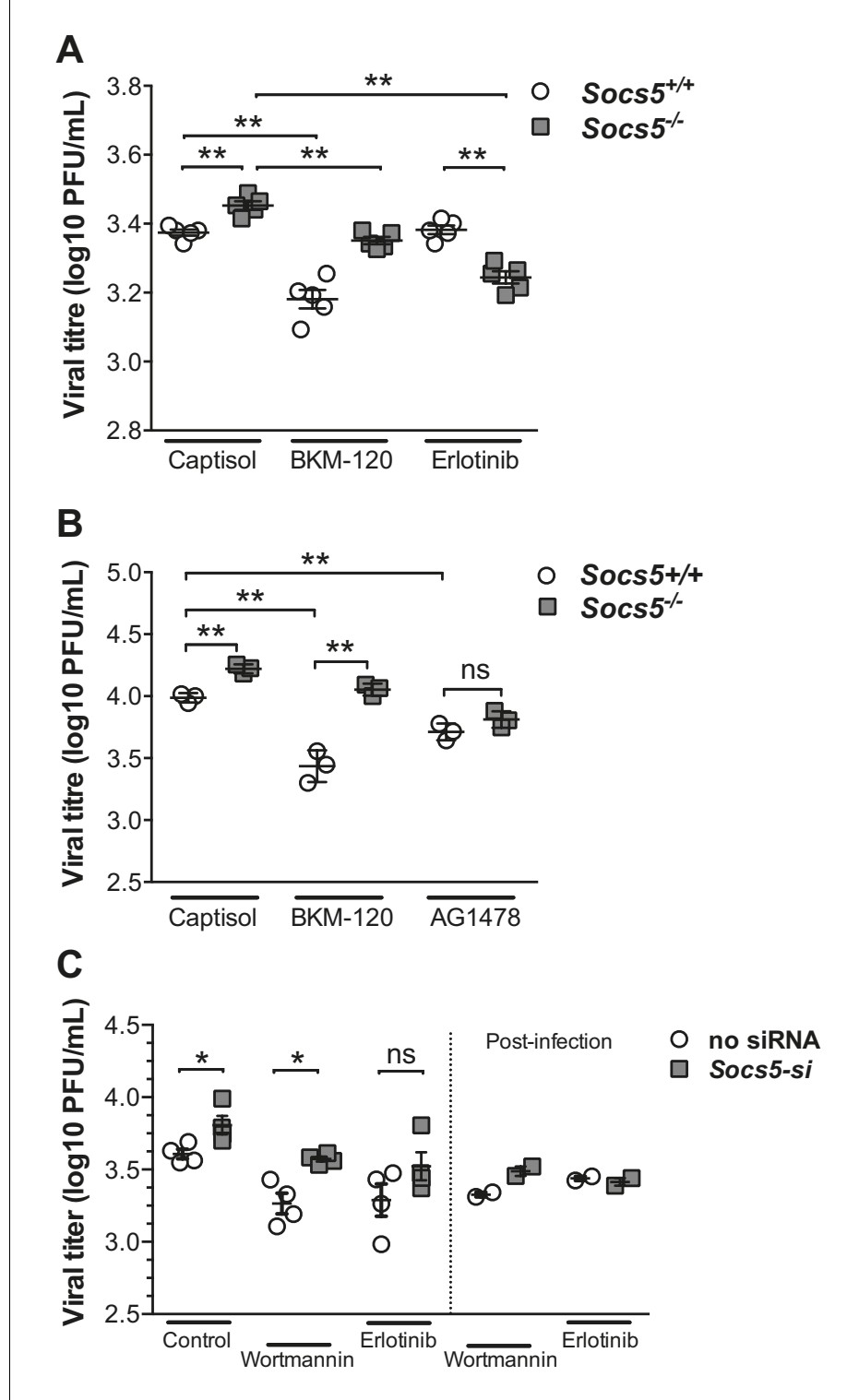

**Figure 7.** SOCS5 protects against influenza virus infection by suppressing virus-induced EGFR activity. (**A**) Mice were infected i.n. with 35 pfu influenza virus H1N1 PR8 and viral titres determined in lung homogenates 24 h post-PR8 infection, by plaque assay. Mice were treated with vehicle (captisol), PI3K inhibitor (BKM-120) or EGFR inhibitor (Erlotinib) 4 h post-PR8 inoculation. (**B**) Primary mouse airway epithelial cells (mAEC) from wild-type and $Socs5^{-/-}$ mice were purified and cultured in vitro for 7 days prior to infection with influenza virus H1N1 PR8 (MOI 1) and then incubated in the presence of trypsin for 24 h. PI3K (BKM-120) and EGFR inhibitors (AG1478) were included in the culture media during and post-infection. Culture supernatants were analyzed for infectious virus by
*Figure 7 continued on next page*

*Figure 7 continued*

plaque assay. (**C**) Primary human airway epithelial cells (hAECs) from healthy individuals were cultured as described and infected with H1N1 strain (A/Auckland/1/2009; MOI 5). PI3K inhibitor (Wortmannin) and EGFR inhibitor (Erlotinib) were added 3 h prior to virus inoculation. Culture supernatants were analyzed for infectious virus by plaque assay 24 h post-infection. *p<0.05, **<0.005. (**A–C**; Mean ± S.E.M of indicated *n*). p values were determined by (**A,B**) unpaired student's t-test or (**C**) a Mann-Whitney U test.

Although PI3K is often activated downstream of EGFR, our data also suggest that PI3K signaling might be targeted by SOCS5 independently of the EGFR. Firstly, PI3K p110α levels were elevated in the lungs of uninfected mice with no apparent corresponding increase in EGFR levels (*Figure 4A*). Secondly, we captured a specific SOCS5/PI3K p85/p110 complex in 293T cells, in the absence of an EGFR interaction (*Figure 5A,B* and *Table 2*). Although the PI3K p110α catalytic subunit was constitutively elevated in *Socs5*$^{-/-}$ mouse lungs, expression of the p85 regulatory PI3K subunit appeared to vary with infection (*Figure 4A,B*). A change in p110:p85 ratio in uninfected cells may prime for enhanced PI3K catalytic activity (*Ueki et al., 2002*) following receptor tyrosine kinase activity or direct activation by the influenza virus NS1 protein. This is further supported by the increased phosphorylation of the PI3K downstream substrate AKT in the lungs of *Socs5*-deficient mice during infection (*Figures 4* and *6E*).

Reduced SOCS5 expression in primary human epithelial cells also resulted in increased viral HA protein within 2 h of infection, indicating that early events in the viral life cycle such as viral entry and trafficking are enhanced. This appears to result from EGFR-driven PI3K activity, as inhibition of either kinase ablated the SOCS5-dependent effects on HA levels (*Figure 6—figure supplement 3*). Activation of the viral detection pathways results in the cleavage and shedding of EGFR ligands such as TGFα (*Ito et al., 2015*). If this were the case, EGFR activity may be enhanced in the absence of SOCS5. Ligand-independent EGFR clustering is thought to preferentially result in autophosphorylation of the PI3K docking site (Tyr 992), but not that of other more distal sites (*Huang et al., 2016*). We speculate that SOCS5 may inhibit virus-induced receptor clustering and autophosphorylation and when SOCS5 is reduced, selective activation of the PI3K sites. Coupled with increased PI3K p110 levels, this would give a net increase in PI3K activity at the membrane, resulting in increased receptor endocytosis (enhanced entry).

Interestingly and somewhat surprisingly, the increase in viral titres observed in *Socs5*$^{-/-}$ mice at 24 h post-inoculation did not resolve with the inhibition of PI3K activity, despite the importance of PI3K activity in this system and the evidence that SOCS5 can independently target PI3K (as discussed above). This suggests that in vivo, SOCS5 regulates PI3K-independent EGFR effects that are required for restraint of the infection. There are many consequences of EGFR signaling, not least of which is a Ras/MAPK-driven transcriptional response, which may explain the difference between the in vivo results and the short-term infections in vitro in primary human cells.

The SOCS proteins are known to function as adaptors for E3 ubiquitin ligases. If SOCS5 regulated phosphorylated EGFR or PI3K p110α by ubiquitination and proteasomal degradation, pEGFR/PI3K levels should increase in the presence of a proteasomal inhibitor such as MG132 and the differences observed with reduced or increased SOCS5 expression should be equalized. However, incubation with MG132 did not change the levels of either target (*Figure 6—figure supplement 4*). Similarly, the differences consistently observed with altered SOCS5 levels, were retained in the presence of MG132 (*Figure 6—figure supplement 4*). Additionally, we were unable to demonstrate SOCS5-mediated ubiquitination of PI3K p110α in an in vitro assay using recombinant proteins (*Figure 6—figure supplement 5*). Collectively, this suggests that SOCS5 does not regulate either pEGFR or PI3K through proteasomal degradation.

While proteasomal degradation would have been a reasonable explanation for the elevated pEGFR and PI3K p110α levels in SOCS5-deficient lungs and cells, these results suggest that SOCS5 has alternate ways to modulate protein function and that its regulation of EGFR signaling is more complex than earlier suppositions (*Nicholson et al., 2005*). SOCS1, SOCS3 and cytokine-inducible SH2-containing (CIS) protein can interact with and inhibit the enzymatic activity of the JAK kinases (*Delconte et al., 2016*; *Linossi et al., 2013a*). SOCS5 belongs to the sub-group of SOCS proteins

with an extended N-terminal region (369 residues). It is likely that the N-terminal region acts as a scaffold to regulate multiple proteins, in addition to its activity as an E3 ligase.

Whilst we don't fully understand the mechanism, SOCS5 may inhibit the ability of EGFR to suppress RIG-I-IFN I production (*Ueki et al., 2013*), possibly through scaffolding larger complexes or targeting an unknown substrate for ubiquitination. Alternatively, SOCS5 may effect a switch in EGFR signaling the result of which is greater viral dependence on EGFR. This is apparent in the presence of the PI3K inhibitor, where even larger differences are observed between SOCS5 wild-type and depleted cells (*Figure 7*), and where the levels of virus in *Socs5⁻ᐟ⁻* cells drop below that of wild-type cells in the presence of EGFR inhibitor (*Figure 7A*).

Although overexpression of SOCS5 in mammalian cells has been shown to downregulate EGF signaling (*Nicholson et al., 2005*), *Socs5⁻ᐟ⁻* mice have no gross EGF-related phenotype (*Brender et al., 2004*). Further, we have cultured *Socs5⁻ᐟ⁻* embryonic fibroblasts and found no differences in EGF-induced signaling (data not shown). Thus there is a unique role for SOCS5 in regulating viral activation of the EGFR in lung epithelial cells and this study confirms an important role for endogenous EGFR activity in facilitating viral infections.

Identifying SOCS5 as a protective regulator of influenza infection and the subsequent hyper-inflammatory response in airway epithelium, offers the potential to target either SOCS5 itself (enhancing SOCS5 expression or using mimetics) or SOCS5 interacting proteins such as PI3K, to reduce the damaging inflammation that is characteristic of severe influenza and chronic respiratory diseases. Similarly, this study also raises the possibility that short-term use of EGFR inhibitors in severe influenza may be beneficial, particularly when coupled with direct delivery to the lung; facilitating rapid action, whilst reducing systemic toxicity.

We have shown SOCS5 to be a critical regulator of the intracellular pathways that control influenza virus infection. Mice lacking SOCS5 have higher viral loads and exhibit a neutrophilic inflammatory response associated with increased morbidity. Importantly, restoring SOCS5 expression in bronchial epithelial cells from COPD patients reduced viral entry, suggesting that targeting SOCS5 and/or its interacting partners may be valid intervention strategies to alleviate the severe pathology associated with influenza virus infection in COPD patients, immunocompromised individuals or infection with highly pathogenic influenza strains. Future studies will attempt to understand why SOCS5 levels are reduced in epithelial cells from COPD patients.

## Materials and methods

### Influenza viruses

Human influenza A/Puerto Rico/8/1934 (H1N1; PR8), A/Auckland/1/2009 (H1N1) A/Wellington/43/2006 (H3N2) and A/Vietnam/1203/04 (H5N1) strains and the avian influenza A/sharp-tailed sandpiper/Australia/6/2004 (H11N9) (*Hurt et al., 2006*) strain were obtained from the WHO Collaborating Centre for Reference and Research on Influenza (Parkville, Victoria, Australia). All work with H5N1 was performed in the Level 4 Containment Facility at the Commonwealth Scientific and Industrial Research Organization (CSIRO), Australian Animal Health laboratory, Geelong, Victoria, Australia. Viral titres were determined by plaque assay on monolayers of Madin Derby canine kidney (MDCK; RRID:CVCL_0422) cells (*Huprikar and Rabinowitz, 1980*).

### Mice

SOCS5-deficient mice (*Socs5⁻ᐟ⁻*) have been described previously (*Brender et al., 2004*) and were backcrossed for 10 generations onto a BALB/c background. *Socs4⁻ᐟ⁻* and *Sirpa⁻ᐟ⁻* Balb/c mice have been described previously (*Kedzierski et al., 2014*; *Legrand et al., 2011*). All mice were bred at the Walter and Eliza Hall Institute animal facility. Animal experiments followed the NHMRC Code of Practice for the Care and Use of Animals for Scientific Purposes guidelines and were approved by the Walter and Eliza Hall Institute's Animal Ethics Committee.

### Virus infection and tissue collection

Mice were lightly anaesthetized by inhalation of methoxyflurane and infected intranasally (i.n.) with 25–50 plaque-forming units (pfu) of PR8 influenza virus. Kinase inhibitors or vehicle were administered 3 h prior to infection with PR8 influenza virus. The EGFR inhibitor Erlotinib (MedKoo

Biosciences) was administered by oral gavage (20 mg/kg in 6% Captisol), the PI3K inhibitor BKM-120 (Active Biochem) by I.P. injection (50 mg/kg in 12% Captisol). Control mice received 12% Captisol (Ligand Pharmaceuticals) by I.P. injection.

Weight was monitored daily as early as from day one post-infection. Mice were sacrificed at various time points and bronchoalveolar lavage (BAL) samples and entire lungs collected for analysis. Lungs were mechanically homogenised using a Polytron System PT 1200 (Kinematica), centrifuged at 836 $g$ for 10 min and the supernatant harvested for detection of infectious virus by plaque assay or for cytokine analysis. Alternatively, lungs were collected in TRIZOL (Invitrogen) and homogenised for Q-PCR analysis or snap-frozen in liquid nitrogen prior to protein analysis.

## Immunophenotypic staining

Cells recovered from BAL samples were stained with antibodies to CD4-PE, B220-APC, CD8-PerCP, CD11b-FITC, CD11c-APC, Ly6C-PerCP (BD Biosciences or BioLegend) and analyzed by flow cytometry on a FACS Canto (BD Biosciences), using FlowJo software (Tree Star). Different combinations of antibodies were used as indicated in the text.

## Bone marrow chimeras

Bone marrow chimeras were established as previously described (*Kedzierski et al., 2014*). Briefly, irradiated recipient mice (BALB/c Thy1.1) were reconstituted by I.V. injection with $3 \times 10^6$ T cell-depleted bone marrow cells from donor mice (Thy1.2 $Socs5^{-/-}$ or BALB/c). Following injection, the mice were allowed to reconstitute for at least eight weeks prior to use, blood samples were collected (by submandibular bleeding), and reconstitution assessed by FACS analysis of Thy 1.2/Thy 1.1 T cells. On average, wild-type mice showed >80% reconstitution with $Socs5^{-/-}$ bone marrow, while $Socs5^{-/-}$ mice showed >70% reconstitution with wild-type bone marrow.

## Immunohistochemistry

BALB/c and $Socs5^{-/-}$ control mice were inoculated with 50 pfu influenza virus PR8 and lungs harvested at day three post-infection. Uninfected lungs were collected as controls. Sections were incubated with SOCS5 polyclonal rabbit antibody (Abcam), followed by biotinylated secondary antibody, and antibody binding visualized using ABC reagent (Fuzhou Maixin Biotechnology) as per the manufacturer's instructions. Slides were counterstained with hematoxylin. *See also* Supplementary information.

## Cytokine analysis

Cytokine levels in BAL were analyzed using the BioPlex Pro Assay (BioRad). IFNα and β were detected in lung homogenates by sandwich ELISA using the following mouse monoclonal capture antibodies, IFNα (clone F18; Thermo Scientific), IFNβ (clone 7F-D3; Abcam), together with biotinylated detection antibodies (PBL Interferon Source). IFNγ$_2$ was detected by ELISA (R and D Systems).

## Preparation of primary mouse airway epithelial cells (mAEC)

Mouse AEC cultures were prepared as described (*Thomas et al., 2014*), with minor modifications. Briefly, lungs were digested in 1.5 mg/mL Pronase (Roche) and 0.1 mg/ml DNase I (Sigma-Aldrich) for 60 min at 37°C in 5% $CO_2$. Single cell suspensions were incubated with purified rat anti-mouse CD45 antibody (BD Biosciences) and epithelial cells negatively enriched using BioMag goat anti-rat Ig-coupled magnetic beads (Qiagen). Flow cytometry was used to confirm cell purity, which was approximately 95% with mouse anti-EpCAM antibodies (BioLegend) and cultures were on average 20% positive for podoplanin (AEC type I) and 70% for CD74 (AEC type II).

## Infection of mAEC

Monolayers of mAEC at approximately 80% confluency, were incubated with the influenza strain PR8 (MOI as indicated) in serum free media for 1 h, washed and then placed in media supplemented with 2% FCS. To examine virus infection, cells were fixed with 80% acetone at 8 h post-infection and stained with anti-nuclear viral protein antibodies, as described (*Tate et al., 2010*). The percentage of infected cells was determined in a minimum of four random fields with at least 200 cells counted per sample. To examine virus replication, monolayers of mAEC were incubated with PR8 (MOI 1) in

**Table 3.** Subject characteristics.

| | Healthy | COPD | Smoker | P – value |
|---|---|---|---|---|
| Number | 5 | 5 | 5 | NA |
| Sex (% Female) | 60% | 60% | 40% | p=0.6 |
| Mean Age (S.D.) | 61 (16.29) | 68 (12.35) | 64.33 (12.82) | p=0.07 |
| Mean FEV$_1$ (S.D.) [*] | 98% (8.61) | 43% (10.63) | 97.5% (10.30) | p<0.001 |
| Cigarette (Packs/year; S.D.) | 0 | 43 (10.66) | 23 (11.54) | p<0.001 |
| Years abstinent (S.D.) | 0 | 12.35 (5.28) | 0 | NA |
| ICS (percent treated) | 0 | Seretide/Tiotropium/Salbutamol (60%) Tiotropium (40%) | 0 | NA |

[*] FEV$_1$ refers to the forced expiratory volume in 1s expressed as a percentage of the predicated value.

The statistical analysis used for this table is ANOVA for multiple groups.

NA = Not applicable.

serum free media for 1 h and then incubated in media supplemented with 2% FCS and 4 µg/mL trypsin. Levels of infectious virus in cell supernatants were determined by plaque assay on MDCK cells.

## Real-time quantitative PCR (Q-PCR)

Real-time Q-PCR was performed essentially as described (*Lee et al., 2009*). Additional primers sequences are as follows: hIFNλ Forward (F): ACAGCTTCAGGCCACAGCAGAGC; Reverse (R): ccaggagtctccttgctctggg, hSOCS5 F: GCCACAGAAATCCCTCAAATTG; R:ggagcatgtcgagagtaggaatct, hRIG-I F: GCCCTCATTATCAGTGAGCA; R: ATCTCATCGAATCCTGCTGC. Relative expression was determined by normalizing the amount of each gene to the housekeeping gene mouse 18s rRNA using the following primer sequences: F: ACGGACCAGAGCGAAAGCAT and R: CGGCATCGTTTA TGGTCGGA or to human hypoxanthine phosphoribosyltransferase (HPRT).

## Antibodies and reagents

The mouse monoclonal antibody to SOCS5 was generated in-house and recognises an epitope in the SOCS5 N-terminal region. Antibodies to phospho-EGFR (Tyr 1068), EGFR, PI3K p85, PI3K p110α, RIG-I, phospho-MAPK (Thr202/204), MAPK, phospho-AKT1 (Ser473), AKT, phospho-STAT3 and STAT3 were obtained from Cell Signaling Technology. Anti-human SOCS5, anti-GAPDH and anti-avian Influenza A hemagglutinin antibodies were obtained from Abcam.

## Immunoprecipitation and immunoblotting

Whole lungs were dounce homogenized in KALB lysis buffer (*Linossi et al., 2013b*). Protein concentrations were determined by the BCA method (Pierce). A549 cells (RRID:CVCL_0023) were treated with 10 µM MG132 (6 h) and pervanadate solution (H$_2$O$_2$/25 µM Na$_3$VO$_4$) (20 min) prior to cell lysis in 1% NP-40 buffer (1% v/v NP-40, 50 mM HEPES, pH 7.4, 150 mM NaCl, 1 mM EDTA, 1 mM NaF, 1 mM Na$_3$VO$_4$). SOCS5 was immunoprecipitated using the in-house anti-SOCS5 antibody conjugated to Sepharose. Alternatively, 293 T cells (RRID:CVCL_0063) were transiently transfected with vector alone or cDNA expressing Flag-tagged mouse SOCS5, using FuGene6 (Promega). 48 h post-transfection, cells were pre-treated with pervanadate solution for 30 min and lysed in 1% NP-40 buffer. Flag-SOCS5 was immunoprecipitated using M2-beads (Sigma) and proteins eluted with 0.5% sodium dodecyl sulfate (SDS) and 5 mM dithiothreitol (DTT), prior to tryptic digest and mass spectrometry. Endogenous SOCS5 was immunoprecipitated using 5 µg in-house antibody and protein-A Sepharose. Wild-type and mutant SOCS5 expression vectors, immunoprecipitation, gel electrophoresis and immunoblotting have been described previously (*Linossi et al., 2013b*). Cell lines were tested as mycoplasma free.

## Trypsin digestion

Equal amounts of lung lysates (~200 µg) or eluates from SOCS5 immunoprecipitates were subjected to tryptic digest using the FASP protein digestion kit (Protein Discovery) (*Wiśniewski et al., 2009*),

with the following modifications. Protein material was reduced with TCEP (5 mM final) and digested overnight with 2 µg sequence-grade modified trypsin Gold (Promega) in 50 mM $NH_4HCO_3$ at 37°C. Peptides were eluted with 50 mM $NH_4HCO_3$ in two 40 µL sequential washes and acidified in 1% formic acid (final). Mass spectrometric analysis was performed as described in Appendix 1.

## Primary human cell culture, viral infection and siRNA depletion

Primary human airway epithelial cells (hAECs) were obtained by endobronchial brushing during fibre-optic bronchoscopy from healthy individuals, individuals who smoked but had no history of lung disease, or individuals who smoked and had chronic obstructive pulmonary disease (COPD). Healthy subjects had no history of smoking or lung disease and had normal lung function. *See Table 3* for patient details. All subjects gave written informed consent and all procedures were performed according to approval from the University of Newcastle Human Ethics Committee. hAEC were cultured as described and used at passage two (*Hsu et al., 2015*). H1N1 (A/Auckland/1/2009), H3N2, H11N9 and H5N1 were diluted in the appropriate serum free media and added to cells at an MOI of 5 (H3N2, H1N1, and H11N9) and 0.005 (H5N1). After 1 h the virus was removed and replaced with serum-free media. Cells were lysed 24 h post-infection and analyzed by Q-PCR or immunoblotting. *Socs5* was depleted using Silencer Select pre-designed siRNAs (Life Technologies), which were reverse transfected into hAECs using siPORT NeoFX transfection agent (Life Technologies), 24 h prior to infection. Wortmannin (100 nM; Sigma-Aldrich) or Erlotinib (100 nM; Selleckchem) was added to hAECs 24 h before infection or 1 h after virus inoculation.

## Statistical analysis

Statistical analysis was performed using an unpaired t-test with a 95% confidence level.

# Acknowledgements

The authors thank Shannon Oliver, Catherine Hay and Liana Mackiewicz from the Walter and Eliza Hall Institute of Medical Research, for excellent animal husbandry. This work was supported in part by the National Health and Medical Research Council (NHMRC), Australia (Project grant #1045762, Program grant #1016647), an NHMRC IRIISS grant 361646 and a Victorian State Government Operational Infrastructure Scheme grant. NAN and PMH are supported by NHMRC fellowships, KK by an NHMRC CDA2 Fellowship, MDT and MRS by an NHMRC Peter Doherty Fellowship, GTB by an Australian Research Council Future Fellowship and EML by an Australian Postgraduate Award. None of the authors has a financial interest related to this work.

# Additional information

### Funding

| Funder | Grant reference number | Author |
|---|---|---|
| Victorian State Government, Australia | Operational Infrastructure Scheme grant | Lukasz Kedzierski<br>Tatiana B Kolesnik<br>Edmond M Linossi<br>Laura Dagley<br>Sarah Freeman<br>Simon M Chatfield<br>Jeffrey J Babon<br>Nicholas Huntington<br>Gabrielle Belz<br>Andrew Webb<br>Nicos A Nicola<br>Sandra E Nicholson<br>Giuseppe Infusini |
| National Health and Medical Research Council | IRIISS grant 361646 | Lukasz Kedzierski<br>Tatiana B Kolesnik<br>Edmond M Linossi<br>Laura Dagley<br>Sarah Freeman<br>Giuseppe Infusini<br>Simon M Chatfield |

| | | Nicholas Huntington |
| | | Gabrielle Belz |
| | | Andrew Webb |
| | | Nicos A Nicola |
| | | Sandra E Nicholson |
| | | Jeffrey J Babon |
| National Health and Medical Research Council | Fellowship | Michelle D Tate |
| | | Malcolm R Starkey |
| | | Nicos A Nicola |
| | | Katherine Kedzierska |
| | | Nicholas Huntington |
| | | Philip M Hansbro |
| Australian Federal Government | Australian Postgraduate Award | Edmond M Linossi |
| National Health and Medical Research Council | Project grant #1045762 | Peter AB Wark |
| | | Philip M Hansbro |
| National Health and Medical Research Council | Program grant #1016647 | Nicos A Nicola |
| | | Sandra E Nicholson |
| Australian Research Council | Future Fellowship | Gabrielle Belz |

The funders had no role in study design, data collection and interpretation, or the decision to submit the work for publication.

## Author contributions

LK, Conceptualization, Formal analysis, Investigation, Methodology, Writing—original draft; MDT, ACH, EML, LD, Formal analysis, Investigation, Methodology, Writing—review and editing; TBK, ZD, Formal analysis, Investigation, Methodology; SF, SMC, Investigation, Methodology; GI, Data curation, Formal analysis; MRS, Conceptualization, Investigation, Performed preliminary QPCR data showing reduction of Socs5 mRNA in an experimental COPD model; NLB, Investigation, Methodology, Writing—review and editing; JJB, AW, PABW, Resources, Supervision, Methodology; NH, Conceptualization, Resources, Provided the Sirpa-/- mice used in Fig 1-Fig supplement 1 and had knowledge and expertise in understanding the defects in these mice, having performed the orginal analysis. It was important that we had a model with a defect in innate immunity for comparison with the Socs5-deficient mice to address the reviewer/s comments; GB, Resources, Methodology, Writing—review and editing; NAN, Resources, Supervision, Funding acquisition, Writing—review and editing; JX, Conceptualization, Resources, Supervision, Methodology, Writing—review and editing; KK, Resources, Supervision, Methodology, Writing—review and editing; PMH, Conceptualization, Resources, Supervision, Funding acquisition, Project administration, Writing—review and editing; SEN, Conceptualization, Supervision, Funding acquisition, Writing—original draft, Project administration, Writing—review and editing

## Author ORCIDs

Michelle D Tate, http://orcid.org/0000-0002-0587-5514
Alan C Hsu, http://orcid.org/0000-0002-6640-0846
Sandra E Nicholson, http://orcid.org/0000-0002-1314-2134

## Ethics

Human subjects: All subjects gave written informed consent and all procedures were performed according to approval from the University of Newcastle Human Ethics Committee (Ethics Number: H-163-1205).

Animal experimentation: Animal experiments followed the NHMRC Code of Practice for the Care and Use of Animals for Scientific Purposes guidelines and were approved by the Walter and Eliza Hall Institute's Animal Ethics Committee (Ethics Number: 2014.029).

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

## Appendix 1

### Additional experimental procedures

#### Immunohistochemistry

Lungs were fixed in 4% paraformaldehyde and embedded in paraffin. For immunohistochemistry, slides were baked at 68°C for 60 min, deparafinized twice in xylene (10 min), followed by rehydration in graded dilutions of ethanol in water (100%, 100%, 95%, 90%, 80%, 70%; 5 min for the first two dilutions then 3 min for subsequent dilutions), and washed in PBS. Antigen retrieval was performed by heating the slides in citrate buffer (pH 6.0) for 20 min at 92°C. Cooled slides were washed twice in PBS and incubated in 3% $H_2O_2$ for 10 min at room temperature (RT) followed by PBS washes. Blocking was performed in 2% bovine serum albumin (BSA) for 20 min followed by overnight staining at 4°C with SOCS5 polyclonal rabbit antibody (Abcam). Slides were washed three times in PBS and incubated with biotinylated secondary antibody and ABC reagent as per the manufacturer's instructions. Slides were counterstained with hematoxylin and mounted in Permount Mounting Medium (Fisher Scientific).

#### Quantification of cell-free dsDNA

Cell-free bronchoalveolar lavage samples were incubated with Proteinase K (Roche) 2.5 mg/mL for 1 h at 37°C on an orbital shaker. PicoGreen (Invitrogen) was used to measure the dsDNA content of the protease-treated solution (*Chen et al., 2007*) according to manufacturer's instructions. dsDNA content was compared to a standard curve of lambda DNA (Invitrogen).

#### Mass spectrometry (MS) and data analysis

Acidified peptide mixtures were analyzed by nanoflow reversed-phase liquid chromatography tandem mass spectrometry (LC-MS/MS) on a nanoAcquity system (Waters) coupled to a Q-Exactive mass spectrometer equipped with a nanoelectrospray ion source for automated MS/MS. (Thermo Fisher Scientific). Peptide mixtures were loaded on a 20 mm trap column with 180 mm inner diameter (nanoAcquity UPLC 2 G-V/MTrap 5 mm Symmetry $C_{18}$) at 1% buffer B, and separated by reverse-phase chromatography using a 150 mm column with 75 mm inner diameter (nanoAcquity UPLC 1.7 mm BEH130 $C_{18}$) on a 60 min linear gradient set at a constant flow rate of 300 nL/min from 100% buffer A (0.1% formic acid, Milli-Q water) to 30% B (0.1% formic acid, acetonitrile). The Q-Exactive was operated in a data-dependent mode, switching automatically between one full-scan and subsequent MS/MS scans of the ten most abundant peaks. The instrument was controlled using Exactive series version 2.1 build 1502 and Xcalibur 3.0. Full-scans (m/z 350–1850) were acquired with a resolution of 70,000 at 200 m/z. The 10 most intense ions were sequentially isolated with a target value of 10000 ions and an isolation width of 2 m/z and fragmented using HCD with normalized collision energy of 19.5 and stepped collision energy of 15%. Maximum ion accumulation times were set to 50 ms for full MS scan and 200 ms for MS/MS. Underfill ratio was set to 5% and dynamic exclusion was enabled and set to 90 s.

Raw files consisting of high-resolution MS/MS spectra were processed with MaxQuant (version 1.5.2.8) for feature detection and protein identification using the Andromeda search engine (*Cox et al., 2011*). Extracted peak lists were searched against the UniProtKB/Swiss-Prot *Mus musculus* and *Influenza A* virus (strain A/Puerto Rico/8/1934 H1N1) databases and a separate reverse decoy database to empirically assess the false discovery rate (FDR) using strict trypsin specificity allowing up to three missed cleavages. The minimum required peptide length was set to seven amino acids.

*Modifications:* Carbamidomethylation of Cys was set as a fixed modification, while N-acetylation of proteins, oxidation of Met, the addition of pyroglutamate (at N-termini Glu and Gln), phosphorylation (Ser, Thr and Tyr), deamidation (Asn, Gln and Arg), were set as variable modifications. Deamidation of arginines was used to identify citrullinated peptides, resulting in a mass increase of 1 Da. The mass tolerance for precursor ions and fragment ions were 20 ppm and 0.5 Da, respectively. The 'match between runs' option in MaxQuant was used to transfer identifications made between runs on the basis of matching precursors with high mass accuracy (*Cox and Mann, 2008*). PSM and protein identifications were filtered using a target-decoy approach at a FDR of 1%. Protein identification was based on a minimum of two unique peptides.

## Quantitative proteomics analysis

Further analysis was performed using a custom pipeline developed in Pipeline Pilot (Biovia) and R, which utilizes the MaxQuant output files allPeptides.txt, peptides.txt and evidence.txt. A feature was defined as the combination of peptide sequence, charge and modification. Features not found in at least half the number of replicates in each group were removed. Proteins identified from hits to the reverse database and proteins with only one unique peptide were also removed. To correct for injection volume variability, feature intensities were normalized by converting to base two logarithms and then multiplying each value by the ratio of maximum median intensity of all replicates over median replicate intensity. Features assigned to the same protein differ in the range of intensity due to their chemico-physical properties and charge state. To further correct for these differences, each intensity value was multiplied by the ratio of the maximum of the median intensities of all features for a protein over the median intensity of the feature. Missing values were imputed using a random normal distribution of values using the mean of the real distribution of values minus 1.8 standard deviations, and a standard deviation of 0.5 times the standard deviation of the distribution of the measured intensities (*Cox et al., 2014*). The probability of differential expression between groups was calculated using the Wilcoxon Rank Sum test excluding any non-unique sequences and any features with modifications other than oxidation and carbamidomethylation. Probability values were corrected for multiple testing using Benjamini–Hochberg method. Cut-off lines with the function $y= -\log10(0.05)+c/(x-x_0)$ (*Keilhauer et al., 2015*) were introduced to identify significantly enriched proteins. c was set to 0.2 while $x_0$ was set to 1 or 2, representing proteins with a 2-fold (log2 protein ratio of 1) or 4-fold (log2 protein ratio of 2) change in protein expression, respectively. The Log2-transformed summed peptide intensities (non-imputed) were visualised in a heat map generated in one-matrix CIMminer, a program developed by the Genomics and Bioinformatics Group (Laboratory of Molecular Pharmacology, Center for Cancer Research, National Cancer Institute).

## In vitro ubiquitination assays

GST-tagged SOCS5 (SH2 domain and SOCS box) in complex with the adapter proteins Elongin B and C was expressed and purified from *E. coli* as previously described (*Linossi et al., 2013b*). Recombinant PI3K (p110α/p85α; V1721) was purchased from Promega. Additional components of the E3 ligase complex (Cullin 5, Rbx2, E2 and E1) were generated as described (*Kershaw et al., 2014*). Full-length, Flag-tagged SOCS5 was expressed and purified from HEK293T cells as previously described (*Linossi et al., 2013b*). Reactions were carried out in 2 mM MgCl$_2$, 2 mM ATP, 25 mM Tris-HCL, pH 7.4 and 1 mM DTT. Ubiquitination reactions were initiated at 37°C by the addition of the E1 enzyme and terminated by the addition of 4x Laemlli reducing sample buffer. Samples were separated by SDS page and transferred to nitrocellulose for immunoblot analysis of PI3K p110α.

## Bronchial epithelial cell line

The minimally immortalised bronchial epithelial cell line (BCi-NS1.1; *Walters et al., 2013*) was cultured in BEBM complete media (Lonza) and was obtained from R.G. Crystal (Weill Cornell Medical College, New York). It was tested as mycoplasma free.

