## [Decision Letter]

[Editors’ note: a previous version of this study was rejected after peer review, but the authors submitted for reconsideration. The first decision letter after peer review is shown below.]

Thank you for submitting your work entitled "Suppressor of Cytokine Signaling (SOCS)5 ameliorates influenza infection via inhibition of EGFR-PI3K signaling" for consideration by *eLife*. Your article has been favorably evaluated by Tadatsugu Taniguchi as the Senior Editor and two reviewers, one of whom is a member of our Board of Reviewing Editors. Our decision has been reached after consultation between the reviewers. Based on these discussions and the individual reviews below, we regret to inform you that your work will not be considered further for publication in *eLife*. Should further experiments allow you to address all of the reviewers’ comments, we would be happy to consider the revised paper as a new submission.

*Reviewer #1:*

Influenza virus is a major pathogen of humans and animals. In this manuscript, the authors report studies of influenza virus infection of SOCS5 knockout mice, cell cultures and biochemical analysis of EGFR-PI3K signaling pathway. They propose that SOCS5 is a critical regulator of influenza virus infection by regulating EGFR-PI3K signaling. The study also suggests the decreased level of SOCS5 gene is responsible for increased susceptibility of individuals with COPD to flu infection. It has been known for some time that SOCS5 can physically associate with EGFR, the activity of EGFR/PI3K and AKT has also been linked to flu infection in previous studies. This is the first study linking all these parts together and it provides some new insights into the host-virus interaction of this important virus. However, evidence provided in the manuscript is only suggestive and cannot support the major claims of the paper.

1) A large amount of studies using gene knockout mice have indicated many genes, for example SOCS4 and innate immunity sensors, are involved in regulating influenza virus infection. The importance of SOCS5 gene in the infection should be compared with at least one or two of them, however, no such data was presented.

2) In addition to EGFR, additional ErbB/EGFR family members were also found to be regulated by SOCS5; moreover, it's known that influenza virus doesn't use a specific protein receptor on the cell surface, but instead binds to sialic acid of many cell surface proteins and glycolipids. It is likely that other receptors engaged with SOCS5 might also contribute to the infection by signaling and/or mediating viral entry.

3) Although an intrinsic epithelial cell defect may be responsible for the phenotype observed in *Socs5^−/−^* mice, no convincing evidence showing viral entry was enhanced.

*Reviewer #2:*

In this study, authors found that *Socs5^−/−^* mice were more sensitive to influenza virus infection than WT mice, and SOCS5 limited influenza virus entry. They also found that SOCS5 elves were lower in epithelial cells from COPD patients, which is consistent with increased susceptibility to influenza. Phenotype analysis of the paper is technically sound and their findings are interesting. However, there is no direct supporting evidence for their main conclusion that "SOCS5 restricts influenza virus entry", and also no explanation for this molecular mechanism.

1) Authors should show convincing data for the regulation of Influenza virus entry by SOCS5. For example, authors should measure binding and nuclear entry of labeled influenza virus into primary epithelial cells from WT and SOCS5-deficient mice as well as cells from COPD patients and SOCS5 siRNA treated cells.

2) This reviewer could not find data showing that influenza virus replication other than HA mRNA is enhanced in human SOCS5-knockdown cells or cells from COPD patients. Viral titration with plaque assay is required.

3) Figure 2 suggested reduced IFNs in KO cells. Are there any effect of IFNs? Is STAT1 lower in KO cells? This may be able to explain higher replication of virus at low MOI. Authors should use anti-IFN antibodies to avoid the effect of IFNs.

4) Figure 7: The mechanisms to suppress viral entry and viral replication by SOCS5 and Akt are not explained. They cited papers, but did not mention mechanisms in the text, and did not show supporting data in their system.

5) Figure 6: "In contrast to infection with H3N2, H1N1 and H11N9, infection with the highly pathogenic avian virus H5N1 dramatically suppressed Socs5 expression relative to the media control (Figure 6)." What is the mechanism? Can H5N1 replicate more than other virus ties in human epithelial cells?

6) How viral infection activates EGFR as shown in Figure 6? Authors should measure levels of EGFR ligands in Figure 2.

[Editors’ note: what now follows is the decision letter after the authors submitted for further consideration.]

Thank you for submitting your article " Suppressor of Cytokine Signaling (SOCS)5 ameliorates influenza infection via inhibition of EGFR signaling " for consideration by eLife. Your article has been favorably evaluated by Wenhui Li as the Senior Editor, a Reviewing Editor, and three reviewers.

The reviewers have discussed the reviews with one another and the Reviewing Editor has drafted this decision to help you prepare a revised submission.

Summary:

The manuscript shows an interesting set of data that identify roles for the role of SOCS5 in restricting influenza A virus in the airway epithelium, with some insights into the possible mechanisms negatively regulating the role of SOCS5 in EGFR kinase activity and PI3K activity. While the data are incremental in that they take forward the directions indicated by recent literature, they do provide important new information well worth publication, if the following issues are addressed.

Essential revisions:

It would be useful to have four sets of concerns addressed substantively.

1) The first concern relates to the mechanisms by which SOCS5 affects EGFR or PI3K activities. Authors should provide more clear biochemical data of SOCS5-mediated EGFR or PI3K suppression. Authors addressed the possibility in the Discussion that SOCS5 inhibits cluster-induced autophosphorylation. The reduction of SOCS5 expression leads to selective activation of the PI3K sites that would give a net increase in PI3K activity at the membrane, resulting in increased receptor endocytosis. However, this possibility is quite speculative. Another possibility may be that the SOCS5 binding to PI3K p110 leads to ubiquitination dependent degradation. This hypothesis is supported by the data of Figure 4, which shows increased protein levels of PI3K p110 in *Socs5^−/−^* cells, and also Figure 5 which showed interaction of SOCS5 with PI3K complex. Authors should show in vitro ubiquitination of p110 by SOCS5 and enhanced proteasomal degradation of p110 by SOCS5 in infected or EGFR activated cells. If second possibility is the case, increase of p110 levels in COPD (Figure 6—figure supplement 4) can be explained easily.

2) The claimed differences in phosphorylation and/or upregulation of several signaling proteins (Figure 6) between healthy persons and COPD patients might be statistically significant, but the question remains whether some of these very weak differences are leading to a real biological difference. Only clear effects are seen when SOCS5 was exogenously overexpressed or silenced, indicating that the decreased induction of SOCS5 by H1N1 in COPD patients has very partial effects on EGFR, PI3K and AKT activity.

3) The observation that both EGFR and PI3K inhibition ablated the SOCS5-dependent effects on virus load and that only EGFR inhibition reversed the increase in virus load both in vivo and in vitro, raises more questions and demands more clarification then the explanation given in the Discussion, which is quite speculative.

4) The final concern relates to the mechanisms of how SOCS5 regulate influenza virus infection via EGFR signaling. This was explained by ligand-independent receptor activation as a consequence of receptor clustering. RIG-I and MDA-5 and TLR3, 7 and 8 are likely to be initial activation signal leading to receptor clustering. It was shown before that Influenza A virus infection induces TGF release and EGFR activation in AECs (Ito et al., 2015). Also, SOCS5 levels were shown to be elevated after EGF stimulation (Kario et al., 2005). However, their explanation was not adequately answered.

---

## [Author Response]

[Editors’ note: the author responses to the first round of peer review follow.]

*Reviewer #1:*

*[…] It has been known for some time that SOCS5 can physically associate with EGFR, the activity of EGFR/PI3K and AKT has also been linked to flu infection in previous studies. This is the first study linking all these parts together and it provides some new insights into the host-virus interaction of this important virus. However, evidence provided in the manuscript is only suggestive and cannot support the major claims of the paper.*

We have now addressed this critique by using EGFR and PI3K inhibitors both in vivo in mice and in ex vivo primary mouse and human epithelial cells. The data obtained in all three systems (in vivo and ex vivo mouse and human) are strikingly consistent and clearly demonstrate that the increased viral load observed in the absence of SOCS5 is due to alterations in EGFR kinase activity and *not to* increased PI3K activity. This data does not negate the role of PI3K in influenza virus infection (and indeed confirms it), nor does it detract from the observation that SOCS5 regulates PI3K signaling. This remains an important novel finding of this study. It specifically addresses the concerns of the reviewer and demonstrates conclusively that SOCS5 restrains influenza virus infection by regulating EGFR kinase activity.

These data have been incorporated as a new figure (Figure 7) with appropriate detail in the Results and Discussion sections.

*1) A large amount of studies using gene knockout mice have indicated many genes, for example SOCS4 and innate immunity sensors, are involved in regulating influenza virus infection. The importance of SOCS5 gene in the infection should be compared with at least one or two of them, however, no such data was presented.*

As requested by the reviewer, we have compared the SOCS5 phenotype with that of mice lacking functional SOCS4. In addition, we also compared the SOCS5 phenotype with that of Rag2^-/-^IL-2Rgc^-/-^ SirpA mice, which lack T/B cells, NK cells and ILCs (Legrand et al., PNAS 2011). We were limited by the availability of gene-targeted mice on the Balbc background, but given that NK cells are a major lymphocytic population in the lung and are important in restraining early viral loads (Stein-Streilein et al., Reg Immunol 1998; J Immunol 1996), we felt that the SirpA mice would be a useful comparison. It is worth noting that very few studies have examined viral loads at 24 h post-inoculation. Loss of SOCS4 function resulted in a significant increase in influenza virus which was greater than that observed with loss of *Socs5*. The increased levels of virus in the *Socs5^-/-^*were comparable to those observed in Rag2^_-/-_^IL-2Rgc^_-/-_^SirpA mice. These data reveal that (i) the Socs5 phenotype is at least as significant as that observed in mice lacking innate lymphocytes, and (ii) SOCS4 appears to have an additional role to that described in Kedzierski et al., PLOS Pathogens 2014. This latter could be due to SOCS4 regulating EGFR, JAK/STAT or TLR2 signaling (Arts et al., J Intern Med 2015; Linossi et al., PLOS One 2013; Bullock et al., Structure 2007) and will be investigated further in a subsequent publication. This data has been added in as Figure 1—figure supplement 1.

*2) In addition to EGFR, additional ErbB/EGFR family members were also found to be regulated by SOCS5; moreover, it's known that influenza virus doesn't use a specific protein receptor on the cell surface, but instead binds to sialic acid of many cell surface proteins and glycolipids. It is likely that other receptors engaged with SOCS5 might also contribute to the infection by signaling and/or mediating viral entry.*

To date, all of the evidence for SOCS5 regulation of EGFR family members is either correlative or derived from overexpression studies. Ours is the first study to conclusively demonstrate a role for SOCS5 in regulating EGFR signaling at the level of endogenous proteins. The requirement for influenza virus binding to sialic acid is undisputed. However, there is substantial evidence supporting a role for EGFR in viral entry and suppression of the anti-viral response (e.g. Eierhoff et al., Plos Pathogens 2010; Ueki et al., JEM 2013). Our new data further support this, with inhibition of EGFR signalling reducing viral loads in primary mouse and human wild-type epithelial cells. Furthermore, using a selective EGFR inhibitor we have now demonstrated that loss of SOCS5 regulation of EGFR activity is responsible for the increased susceptibility of *Socs5^_-/-_^*mice to influenza virus. We agree that SOCS5 may also regulate other receptors, which might be involved in viral infection, but while receptors such as c-Met have been implicated, there is currently no data to suggest that they are regulated by SOCS5.

*3) Although an intrinsic epithelial cell defect may be responsible for the phenotype observed in Socs5^−/−^ mice, no convincing evidence showing viral entry was enhanced.*

Our in vitro human data show differences in viral HA expression with SOCS5-depletion within 2 h of infection and this was normalised by either EGFR or PI3K inhibition (Figure 6, new Figure 6—figure supplement 3). Differences at this timepoint are strongly supportive of a change in viral entry and/or trafficking, particularly in the context of known roles for EGFR-PI3K in viral infection. As a preliminary step towards investigating this, we compared the viral loads by adding inhibitors before or after addition of virus inoculum (plaque assay, 24 h post-infection). Our logic being that if the differences resulted from increased viral entry/trafficking then when the EGFR inhibitor Erlotinib was added post-infection we should retain the increased viral replication in Socs5-deficient cells. If the differences occurred during replication, then addition of EGFR inhibitor post-infection should have the same effect as that observed when added before infection (loss of the Socs5 difference). The latter occurred (see Figure 8), suggesting that the critical effect of Socs5 loss was *not* on viral entry.

Author response image 1.Primary hAECs- infected with H1N1.Left-hand-side of graph: EGFR inhibitor (Erlotinib) and P13K inhibitor (Wortmannin) were added prior to influenza virus infection. Right-hand-side of graph: inhibitors were added after 1 h infection with H1N1 virus. Viral titres were determined by plaque assay, 24 h post-infection.**DOI:**
http://dx.doi.org/10.7554/eLife.20444.021

There is still a consistent increase in HA protein within 2 h of infection in Socs5-depleted cells, which is lost with inhibition of either PI3K or EGFR (new Figure 6—figure supplement 3). However, while this may reflect a modest increase in viral entry/trafficking, the significant difference in viral loads observed at 24 post-infection appear to be due to another EGFR-driven (and PI3K-independent) mechanism. We appreciate that this is not definitive and would certainly like to investigate this result in the future. We have re-worded the manuscript accordingly, referring to early viral infection rather than viral entry. We have also expanded the discussion around possible mechanisms of action.

*Reviewer #2:*

*In this study, authors found that Socs5^−/−^ mice were more sensitive to influenza virus infection than WT mice, and SOCS5 limited influenza virus entry. They also found that SOCS5 levels were lower in epithelial cells from COPD patients, which is consistent with increased susceptibility to influenza. Phenotype analysis of the paper is technically sound and their findings are interesting. However, there is no direct supporting evidence for their main conclusion that "SOCS5 restricts influenza virus entry", and also no explanation for this molecular mechanism.*

Please refer to the response comments for reviewer 1 (point #3).

*1) Authors should show convincing data for the regulation of Influenza virus entry by SOCS5. For example, authors should measure binding and nuclear entry of labeled influenza virus into primary epithelial cells from WT and SOCS5-deficient mice as well as cells from COPD patients and SOCS5 siRNA treated cells.*

Please refer to the response comments for reviewer 1 (point #3).

*2) This reviewer could not find data showing that influenza virus replication other than HA mRNA is enhanced in human SOCS5-knockdown cells or cells from COPD patients. Viral titration with plaque assay is required.*

We have now included plaque assay results from primary human epithelial cells at 24 post-infection showing increased virus production by cells with SOCS5-knockdown (Figure 7).

*3) Figure 2 suggested reduced IFNs in KO cells. Are there any effect of IFNs? Is STAT1 lower in KO cells? This may be able to explain higher replication of virus at low MOI. Authors should use anti-IFN antibodies to avoid the effect of IFNs.*

We were unable to detect phosphorylated STAT1 in the lungs at day 1 post-inoculation. Levels of total STAT1 were unchanged.

*4) Figure 7: The mechanisms to suppress viral entry and viral replication by SOCS5 and Akt are not explained. They cited papers, but did not mention mechanisms in the text, and did not show supporting data in their system.*

Our in vivo and in vitro experiments show that the enhanced susceptibility to influenza infection in *Socs5^-/-^*mice is due to EGFR signaling (Figure 7). How this might impact on viral infection is now included in an extended discussion.

*5) Figure 6: "In contrast to infection with H3N2, H1N1 and H11N9, infection with the highly pathogenic avian virus H5N1 dramatically suppressed Socs5 expression relative to the media control (Figure 6)." What is the mechanism? Can H5N1 replicate more than other virus ties in human epithelial cells?*

Understanding how *Socs5* expression is suppressed is an interesting question, albeit outside the scope of the current manuscript. While we don’t understand the mechanism by which H5N1 results in dramatic suppression of *Socs5* expression, we have previously shown that H5N1 (MOI = 0.005) replicated to 10^7^ pfu/mL compared with H3N2 (10^3^ pfu/mL, MOI = 5) in pBECs (Hsu et al., PONE 2012). The H5N1 work requires a BSL 3/4 containment facility, to which we currently have limited access.

*6) How viral infection activates EGFR as shown in Figure 6? Authors should measure levels of EGFR ligands in Figure 2.*

Viral activation of EGFR (phosphorylation Figure 6) is likely to be ligand-independent receptor activation, which occurs through receptor clustering. In other experiments, we have assessed EGF-ligand activation of EGFR signalling in Socs5-deficient cells (data not shown). This response is robust and not different in the absence of SOCS5.

[Editors' note: the author responses to the re-review follow.]

*Essential revisions:*

*It would be useful to have four sets of concerns addressed substantively.*

*1) The first concern relates to the mechanisms by which SOCS5 affects EGFR or PI3K activities. Authors should provide more clear biochemical data of SOCS5-mediated EGFR or PI3K suppression. Authors addressed the possibility in the Discussion that SOCS5 inhibits cluster-induced autophosphorylation. The reduction of SOCS5 expression leads to selective activation of the PI3K sites that would give a net increase in PI3K activity at the membrane, resulting in increased receptor endocytosis. However, this possibility is quite speculative. Another possibility may be that the SOCS5 binding to PI3K p110 leads to ubiquitination dependent degradation. This hypothesis is supported by the data of Figure 4, which shows increased protein levels of PI3K p110 in Socs5^−/−^ cells, and also Figure 5 which showed interaction of SOCS5 with PI3K complex. Authors should show in vitro ubiquitination of p110 by SOCS5 and enhanced proteasomal degradation of p110 by SOCS5 in infected or EGFR activated cells. If second possibility is the case, increase of p110 levels in COPD (Figure 6—figure supplement 4) can be explained easily.*

We agree that examining whether SOCS5 regulates p110 by ubiquitination and proteasomal degradation is an important point to address. We took several approaches to investigate this:

i) We reduced (siRNA) or increased (exogenous) expression of SOCS5 in combination with a proteasomal inhibitor (MG132) and examined pEGFR, pAKT and PI3K p110α levels in a minimally immortalised bronchial epithelial cell line (BCi-NS1.1). If SOCS5 regulated these targets by proteasomal degradation, we would predict that target levels would increase in the presence of MG132 and that differences observed with knockdown or increased SOCS5 expression would be equalized in the presence of MG132. Consistent with all of our previous data, siRNA knockdown of SOCS5 resulted in increased phosphorylation of EGFR (pEGFR) and AKT (pAKT), increased PI3K p110α levels and increased viral load (HA), whilst constitutive expression of SOCS5 had the converse effect, resulting in reduced pEGFR, pAKT, p110 and HA. However, incubation with MG132 for 4 h did not alter the level of the targets, suggesting that at the endogenous level, degradation through the proteasome is not a major source of regulation. Similarly, the differences observed with either SOCS5 knockdown or expression did not significantly change in the presence of MG132. These results show a consistent phenotype resulting from changes in SOCS5 in a human bronchoepithelial cell line (now consistent across primary bronchial human and mouse epithelial cells, and a human cell line) and the results have been included as part of the Discussion (eighth paragraph) and in the supplementary data (Figure 6—figure supplement 4).

ii) Our second approach utilised an in vitro ubiquitination assay. This assay incorporated recombinant SOCS5-SH2-SOCS box protein produced as a trimeric protein with the adaptors Elongin B/C for stability, and the necessary proteins involved in ubiquitination (recombinant E1, E2, Rbx2, Cullin-5; as in Delconte et al., Nat Immunol 2016). Recombinant PI3K p110α was also added (Promega). These results were negative and we did not observe SOCS5-mediated ubiquitination of PI3K p110α (See Figure 9).

Author response image 2.SOCS5 does not ubiquitinate PI3K p110α.(**A**) Recombinant PI3K p110α (Promega) was incubated with the SOCS5-E3 ligase complex (GST-SOCS5-SH2-BC with Cullin5 and Rbx2) or GST alone as a control, together with ubiquitin, E1 and E2 enzymes at 37°C for the times indicated. Anti- PI3K p110α immunoblot of samples following in vitroubiquitination reaction, with longer and shorter exposures (exp). (**B**) Ponceau stain showing the protein input into each reaction. (**C**) Coomassie stain of all recombinant purified proteins. Recombinant SOCS5-SH2 domain was engineered to contain an N-terminal GST-tag and included the SOCS box sequences for increased stability and solubility when expressed as a ternary complex with elongins B and C, as described (Linossi et al., PLOS One 2013). Note that in the absence of SOCS5, some ubiquitination of PI3K p110α was observed indicating that all components were functional. Addition of SOCS5 blocked this reaction, with no SOCS5-mediated ubiquitination of PI3K p110α observed.**DOI:**
http://dx.doi.org/10.7554/eLife.20444.022

Finally, we modified this assay by using full-length Flag-tagged SOCS5 (produced in 293T cells) instead of recombinant SOCS5-SH2-SOCS box. SOCS5 was affinity purified using anti-Flag antibodies, eluted with Flag peptide, concentrated and added to the in vitro components indicated above. Again, we did not observe SOCS5-mediated ubiquitination of PI3K p110α (see Figure 10).

Author response image 3.Full-length SOCS5 does not ubiquitinate PI3K p110α.(**A**) Recombinant PI3K p110α (+PI3K; Promega) was incubated with either full-length Flag-SOCS5 (WT) or Flag-SOCS5 in which the Elongin C binding site was mutated to prevent E3 activity (mSB), together with Cullin5, Rbx2, ubiquitin, E1 and E2 enzymes at 37°C for 1 h. Anti-PI3K p110α immunoblot of proteins following the in vitro ubiquitination reaction. (**B**) Anti-Flag immunoblot shows levels of SOCS5 when expressed in 293T cells.**DOI:**
http://dx.doi.org/10.7554/eLife.20444.023

Collectively, these results indicate that neither phosphorylated EGFR nor PI3K p110α are targeted by SOCS5 for proteasomal degradation. While this would have been a ‘nice’ explanation for the elevated PI3K p110α levels, these results suggest that SOCS5 uses other ways to modulate protein function, such as by altering the stability of the target protein (either by direct interaction or by ubiquitination). How this occurs is an interesting question and worthy of further investigation, but beyond the scope of the current manuscript. Discussion of these results has been included in the eighth paragraph of the Discussion as “data not shown”; Figure 9 and Figure 10 are not currently included in the manuscript. We are willing to include these figures in supplementary material (currently 7 supplementary figures) if the editors think that this would be a useful addition.

*2) The claimed differences in phosphorylation and/or upregulation of several signaling proteins (Figure 6) between healthy persons and COPD patients might be statistically significant, but the question remains whether some of these very weak differences are leading to a real biological difference. Only clear effects are seen when SOCS5 was exogenously overexpressed or silenced, indicating that the decreased induction of SOCS5 by H1N1 in COPD patients has very partial effects on EGFR, PI3K and AKT activity.*

Some of these differences may be small in AECs, but we think of this in the context of the whole airways. If these small differences occur across a large part of the whole airways in COPD, then they will become more susceptible to influenza virus infection, as will the patient. Further evidence for this is provided by the observation that *Socs5^-/-^*mice are more susceptible to influenza viral infection. There is a fine balance between resistance and susceptibility to infection and this could tip the balance towards infection, especially when combined with other immune defects in COPD. Collectively, the defects in EGFR, PI3K and AKT will be compounded when they occur together, compared to each in isolation. These are typical effects seen in AECs (Hsu et al., AJRCCM 2015).

Another example that we can point to is the modest differences seen in IFN signalling in SOCS1 knockout cells (Brysha et al., JBC 2001; Wormald et al., JBC 2006). While these differences are small, in vivo the consequences are devastating, SOCS1-deficient mice die of an IFN-driven inflammatory disease shortly after birth (Alexander et al., Cell 1999) and similarly *Socs1^-/-^IFNγ^-/-^*mice die rapidly when injected with IFNγ (Brysha et al., JBC 2001).

It is also worth reiterating that these results are very consistent and the effects of reducing *Socs5* expression have now been confirmed in a human cell line, in primary human broncho-epithelial cells (Figure 6—figure supplement 4), and in mice.

We have extended the discussion around this point: “While the individual differences observed in AECs may appear modest, the defects in EGFR, PI3K and AKT need to be considered in combination and importantly, in the context of the entire lung. […] While these differences appear small, in vivo the consequences are devastating, SOCS1-deficient mice die of an IFN-driven inflammatory disease shortly after birth (Alexander et al., 1999) and similarly *Socs1^-/-^IFN*γ^*-/-*^mice die rapidly when injected with IFNγ (Brysha et al., 2001).”

*3) The observation that both EGFR and PI3K inhibition ablated the SOCS5-dependent effects on virus load and that only EGFR inhibition reversed the increase in virus load both in vivo and in vitro, raises more questions and demands more clarification then the explanation given in the Discussion which is quite speculative.*

These data highlight the complexity of anti-viral responses. They show that both EGFR and PI3K/AKT are important in the *Socs5*-dependent effects on viral replication. Essentially, we have examined this in vitro at an early point in viral internalization (2 h post-infection, prior to viral replication) and in vivo at a later timepoint (24 h, prior to adaptive immune response). These results show that EGFR signalling is important in both instances and is responsible for the elevated viral load observed in *Socs5*-deficient mice, whilst SOCS5 regulation of PI3K signalling is only critical during the early phase of viral infection. We agree that these data do raise more questions, and that the explanation given is quite speculative. However, we also consider these to be complex events, which can’t be simplified to provide an easy answer.

Accordingly, we have revised and broadened the discussion around this point: “Interestingly and somewhat surprisingly, the increase in viral titres observed with in *Socs5^-/-^*mice at 24 h post-inoculation did not resolve with the inhibition of PI3K activity, despite the importance of PI3K activity in this system and the evidence that SOCS5 can independently target PI3K (as discussed above). […] It is likely that the N-terminal region acts as a scaffold to regulate multiple proteins, in addition to its activity as an E3 ligase.”

*4) The final concern relates to the mechanisms of how SOCS5 regulate influenza virus infection via EGFR signaling. This was explained by ligand-independent receptor activation as a consequence of receptor clustering. RIG-I and MDA-5 and TLR3, 7 and 8 are likely to be initial activation signal leading to receptor clustering. It was shown before that Influenza A virus infection induces TGF release and EGFR activation in AECs (Ito et al., 2015). Also, SOCS5 levels were shown to be elevated after EGF stimulation (Kario et al., 2005). However, their explanation was not adequately answered.*

We thank the reviewer for bringing this point to our attention. Ito et al., AJPLCMP 2015, showed that A/PR/8 infection of primary human AECs resulted in TGFα release and activation of EGFR signalling. Pro-TGFα cleavage is thought to occur via a TLR/ROS/ADAM protease cascade (Koff et al., AJPLCMP 2008). It is possible that a similar mechanism occurs here and that RIG/TLR activation results in the shedding of EGFR ligands. However, it seems unlikely that this would occur within 2 h of viral infection (as in our human in vitroassays), with the majority of the published data examined at 24 h post-infection.

We have extended our discussion in consideration of this point as follows: “Activation of the viral detection pathways results in the cleavage and shedding of EGFR ligands such as TGFα (Ito et al., 2015). […] We speculate that SOCS5 may inhibit virus-induced receptor clustering and autophosphorylation and when SOCS5 is reduced, selective activation of the P13K sites.”

Shortly after the paper by Kario et al., JBC 2005, was published showing EGF-induction of SOCS5 expression, we attempted but were unable to reproduce the data showing upregulation of SOCS5 in HeLa cells (Nicholson unpublished). However, this does not exclude the possibility that EGF ligands are inducing SOCS5 expression in epithelial cells in vivo, in response to virus.